# LHP1-mediated epigenetic buffering of subgenome diversity and defense responses confers genome plasticity and adaptability in allopolyploid wheat

Zijuan Li[1,2,3,16], Yuyun Zhang[1,2,3,16], Ci-Hang Ding[1,4,16], Yan Chen[5,16], Haoyu Wang[1,2,6,16], Jinyu Zhang [1,2,3], Songbei Ying [1], Meiyue Wang[1], Rongzhi Zhang[7,8,9], Jinyi Liu[1,2,3], Yilin Xie[1,2,3], Tengfei Tang[1,2,6], Huishan Diao[1], Luhuan Ye[2], Yili Zhuang[2], Wan Teng[3,10], Bo Zhang[11], Lin Huang[12], Yiping Tong[3,10], Wenli Zhang [13], Genying Li[7,8,9], Moussa Benhamed [14,15] ✉, Zhicheng Dong [5] ✉, Jin-Ying Gou [4] ✉ & Yijing Zhang [1] ✉

Polyploidization is a major driver of genome diversification and environmental adaptation. However, the merger of different genomes may result in genomic conflicts, raising a major question regarding how genetic diversity is interpreted and regulated to enable environmental plasticity. By analyzing the genome-wide binding of 191 trans-factors in allopolyploid wheat, we identified like heterochromatin protein 1 (LHP1) as a master regulator of subgenome-diversified genes. Transcriptomic and epigenomic analyses of *LHP1* mutants reveal its role in buffering the expression of subgenome-diversified defense genes by controlling H3K27me3 homeostasis. Stripe rust infection releases latent subgenomic variations by eliminating H3K27me3-related repression. The simultaneous inactivation of *LHP1* homoeologs by CRISPR–Cas9 confers robust stripe rust resistance in wheat seedlings. The conditional repression of subgenome-diversified defenses ensures developmental plasticity to external changes, while also promoting neutral-to-non-neutral selection transitions and adaptive evolution. These findings establish an LHP1-mediated buffering system at the intersection of genotypes, environments, and phenotypes in polyploid wheat. Manipulating the epigenetic buffering capacity offers a tool to harness cryptic subgenomic variations for crop improvement.

Polyploidization, which is a recurring event during evolution, is a major driver of genome diversification that promotes adaptive evolution[1–3]. It is pervasive in both plants and animals. Many eukaryotic genomes have a polyploid ancestry, and a substantial proportion of crops underwent recent polyploidization events[4,5]. The rise of allohexaploid wheat (*Triticum aestivum*, 2*n* = 6*x* = AABBDD) approximately 10,000 years ago shaped the evolution of modern humans[6]. However, the convergence of different genomes may not necessarily result in heterosis in polyploids. Instead, the genetic heterogeneity between subgenomes (hereafter referred to as 'subgenome diversity') may result in genetic conflicts. Typical examples include competition between parental genomes and the rapid loss or repression of homoeologous gene

A full list of affiliations appears at the end of the paper. ✉e-mail: moussa.benhamed@universite-paris-saclay.fr; zc_dong@gzhu.edu.cn; jygou@cau.edu.cn; zhangyijing@fudan.edu.cn

copies (biased fractionation or repression)[7–10]. Persistent aneuploidy and decreased fertility are generally associated with nascent allohexaploid plants[11,12]. How sequence diversity in polyploids is effectively interpreted and regulated to enhance environmental plasticity is an important but under-explored issue.

Genetic diversity and environmental stimuli respectively represent internal and external stresses affecting organisms. Phenotypes need to be robust in response to internal and external changes, which requires a buffering system to ensure developmental flexibility ('canalization')[13,14]. Exposure to long-term environmental stress can reveal genetic variants, which may be achieved by repressing buffering systems. These uncovered variants mediating adaptive phenotypes are selected and genetically fixed (assimilation)[15]. In eukaryotes, chromatin and the associated epigenetic mechanisms represent a typical buffering system that stabilizes transcription and cellular homeostasis against internal and external changes, while also reprogramming the transcriptome in response to developmental or environmental cues. Incorporating diverse genomes into a single nucleus yields large-scale epigenomic changes. Epigenetic diversity across subgenomes and the flexible interaction with the transcriptional machinery may provide polyploid wheat with a selective advantage[16–20]. Recent advances detected wide-spread subgenome-unbalanced transcription[21,22], which is mediated by the interplay between genetic and epigenetic diversity, including open chromatin[23,24], histone modifications[25,26], and specific transcription factor (TF) binding densities[27], which may contribute to phenotypic adaptation. However, the available evidence is mostly based on statistical associations between high-throughput data, and a causal relationship between epigenetic programming and phenotypic plasticity remains to be determined.

In addition to influencing transcriptional diversity[17,18], recent investigations on *Arabidopsis*[28] and fungal populations[29,30] indicated that epigenetic architecture directly affects sequence variation and adaptation rates. This effect might be conserved in other eukaryotes, potentially influencing the pace and nature of evolution. Identifying the factors mediating genetic and epigenetic interactions is critical for characterizing the mechanisms by which subgenome diversity and epigenetic plasticity cooperatively promote environmental adaptation and for clarifying and harnessing polyploid plasticity.

In this study, by analyzing subgenomic variations and regulomics data as well as performing genetic screening and functional validation experiments, we revealed how the epigenetic system mediates genotype–environment interactions that determine polyploid wheat phenotypes. The buffering effect of this system on subgenome diversity contributes to developmental plasticity in response to external stimuli and promotes the mutation-selection-fixation cycle of adaptive evolution.

## Results

### LHP1 is a master repressor of subgenome-diversified genes

Polyploid evolution depends on subgenome diversity[31,32]. Identifying the factors regulating subgenome variations is vital for elucidating the causal mechanism underlying polyploid adaptation and evolution. Among the 107,891 high-confidence full-length genome-wide gene models in common wheat, approximately one-third were single copy (sc)-triads (1:1:1 correspondence across subgenomes) (Fig. 1a), whereas the rest were non-sc-homoeologs (varying homoeolog numbers). To reveal the main regulators of non-sc-homoeologs, we collected and generated chromatin immunoprecipitation sequencing (ChIP-seq) and DNA affinity purification sequencing (DAP-seq) data for 191 genome-wide DNA- or chromosome-binding factors and then searched for factors that preferentially regulate non-sc-homoeologs (Supplementary Data 1). On the basis of LASSO regression[33], which is a supervised regularization method used in machine learning because of its powerful built-in feature-selection capability (Fig. 1b), LHP1 was identified as the primary trans-factor regulating non-sc-homoeologs (Fig. 1c,

top), especially the homoeologs absent in one subgenome and multicopies in other subgenome(s), including N:0:N, 0:N:N, 0:1:N (N indicates a minimum of one additional paralog per respective subgenome) (Fig. 1d). Increases in the copy number coincided with increases in the degree of LHP1 binding (Supplementary Fig. 1). The AP2-type TFs were predicted to be among the major regulators of sc-triads (Fig. 1c, bottom). Consistent with these analyses, a large proportion of these enriched homoeolog groups were targeted by LHP1, whereas only 6.8% sc-triads are occupied by LHP1 (Fig. 1e).

In plants, LHP1 is a member of the Polycomb group proteins (PcGs)[34] and is responsible for recognizing and spreading the repressive H3K27me3 mark[35–38]. The specific recruitment of LHP1 is critical for plant development[39–43]. We fine-mapped LHP1-binding sites throughout the common wheat genome on the basis of a ChIP-seq analysis. These binding sites were typically co-localized with the H3K27me3 mark (Fig. 1f), as illustrated by the genomic tracks around *VERNALIZATION1* (*VRN1*) (Fig. 1g). A phylostratigraphic analysis[44], which clustered the genes of common wheat based on their orthologous relationships with nine other species (see "Methods"), revealed an apparent burst of H3K27me3 sites in Triticeae (Fig. 1h), suggestive of an increase in LHP1-mediated epigenetic regulation in Triticeae.

To functionally validate LHP1 and explore the mechanism by which it controls subgenome-diversified genes, we inactivated all three *LHP1* homoeologs in the common wheat variety 'JW1' using the CRISPR–Cas9 editing system and performed ChIP-seq and RNA-seq analyses to assess the effect of a loss-of-function mutation to LHP1 on the expression and epigenetic status of subgenome-diversified genes. Two homozygous triple mutant (*Talhp1-abd*) lines with different frame-shift mutations in all three homoeologs were developed (Fig. 2a and Supplementary Fig. 2). There were apparently fewer H3K27me3 marks in the triple mutant than in the single and double mutants (Supplementary Fig. 3). Similar to the corresponding *Arabidopsis* mutants, the mutants lacking *LHP1* in all three subgenomes flowered early and exhibited dwarfism at maturity (Fig. 2b). We quantified the H3K27me3 levels in wild-type and mutant plants using a modified spike-in method that involved the addition of the same amount of *Arabidopsis* chromatin before ChIP to normalize samples[45] (details are provided in the "Methods" section). The analysis of the *Talhp1-abd* mutant detected a global decrease in H3K27me3 surrounding genes, most of which were non-sc-homoeologs (Fig. 2c, d). The comparison with the transcriptomic changes in *Talhp1-abd* revealed a close association between the decreased H3K27me3 levels and the increased target gene expression levels (Fig. 2e). Considered together, these observations showed that LHP1, through its capacity to control the H3K27me3 level, is a master repressor of subgenome-diversified gene activity.

### LHP1-mediated H3K27me3 is involved in maintaining and promoting genetic diversity across subgenomes

There is accumulating evidence that the evolutionary rate of a gene is predominantly influenced by its expression level rather than its functional importance[46]. The negative correlation between gene expression and the evolutionary rate exists in all three domains of life[46]. The strength and breadth of the effects of LHP1 on the repression of subgenome-diversified gene activity suggest that LHP1 may influence evolutionary processes. We examined the relationship between LHP1 binding and target sequence variations at the population level.

The subgenome diversity in common wheat has two major sources: captured from diploid progenitors (i.e., variations already present across diploids) and newly generated in polyploid populations[31,32]. We first compared the extent of the captured variations (reflected by the differences between subgenomes of the same individual) and polyploidy-generated diversity (reflected by the population diversity π of the same subgenome; i.e., genetic diversity across hexaploid individuals[47]) in a pairwise framework (Fig. 3a). There is a high correlation between captured diversity and

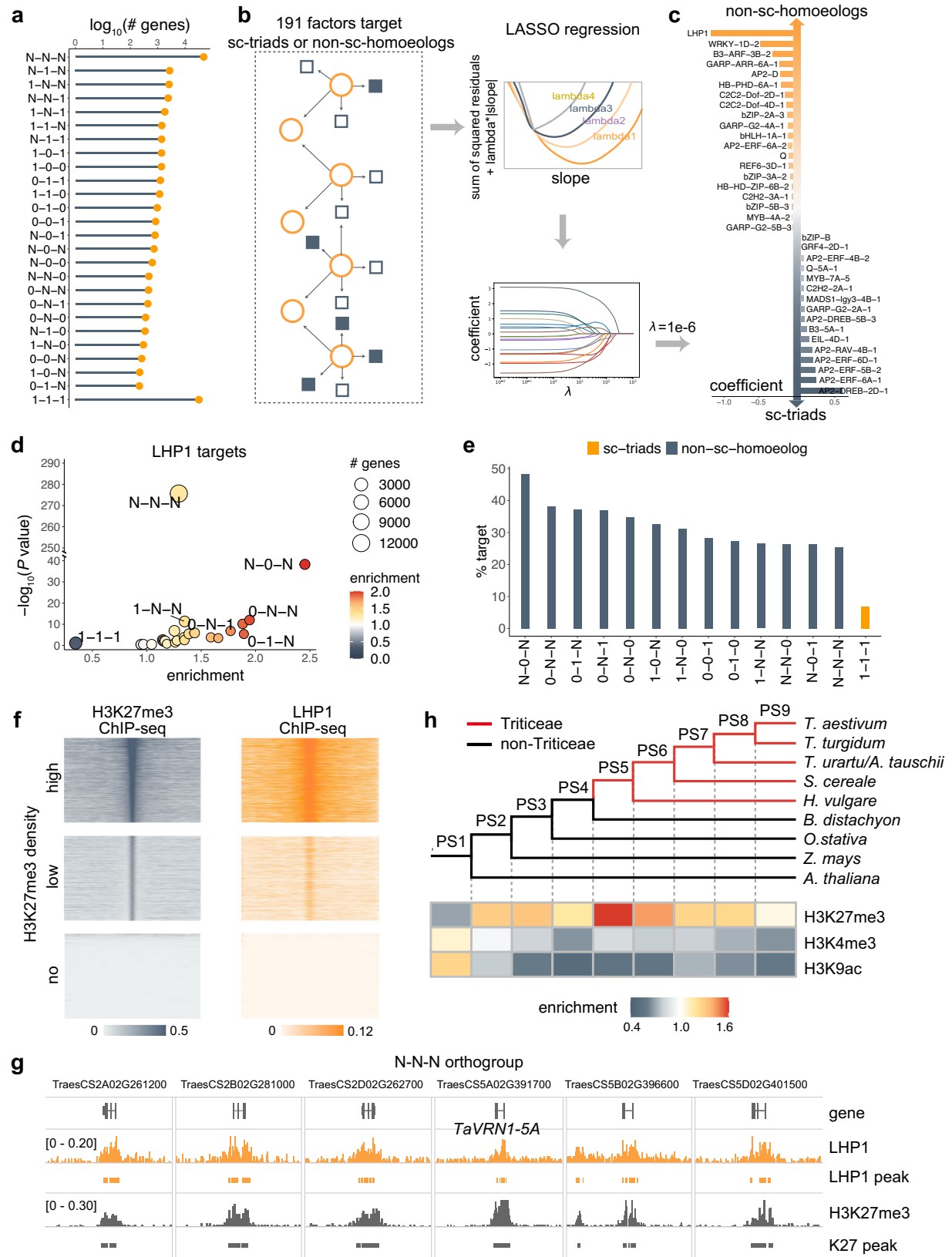

**f** H3K27me3 ChIP-seq / LHP1 ChIP-seq

**g** N-N-N orthogroup

TraesCS2A02G261200 TraesCS2B02G281000 TraesCS2D02G262700 TraesCS5A02G391700 TraesCS5B02G396600 TraesCS5D02G401500

*TaVRN1-5A*

gene / LHP1 / LHP1 peak / H3K27me3 / K27 peak

polyploidy-generated diversity. The variable genome regions in progenitors tended to further diverge after allopolyploidization (region 2 in Fig. 3a), representing selective neutral or weakly deleterious loci likely due to genetic drift.

Next, the enrichment of LHP1 target loci in the above pairwise comparisons was determined. The LHP1-binding sites were highly enriched in regions with high levels of captured diversity, which were further diversified in polyploid populations (Fig. 3b). Among the various epigenetic marks, H3K27me3 was significantly over-represented in these loci (Fig. 3c). Accordingly, LHP1-mediated H3K27me3 repression appears to be closely associated with an increase in subgenomic diversity at the population level, possibly because it promotes neutral

**Fig. 1 | Prediction of the core trans-factors regulating subgenome-diversified genes. a** Number of sc-triads (1:1:1) and non-sc-homoeologs with varying number of homologs across subgenomes in common wheat. **b** Workflow for predicting the core factors regulating sc-triads (1:1:1) and non-sc-homoeologs across subgenomes. **c** LASSO coefficients of the trans-factor binding-related regulation of non-sc-homoeologs (top) and sc-triad (bottom) genes. The coefficient value represents the relative importance of the trans-factor contributing to regulation. **d** Enrichment of different homoeologous groups among LHP1-targeted genes. **e** Fractions of homoeolog groups targeted by LHP1. **f** ChIP-seq read densities

of LHP1 and H3K27me3 surrounding H3K27me3 peaks. **g** Genomic tracks illustrating the co-occupation of LHP1 and H3K27me3 ChIP-seq signals surrounding *VRN1-5A* and its homoeologs. **h** Enrichment of the typical epigenetic marks in eight phylostrata corresponding to the phylogenetic internodes, which are bordered by vertical grids and denote sets of genes whose founder genes originated in the corresponding evolutionary periods (e.g., the left-most phylostratum represents the common ancestor of dicots and monocots). The color represents the enrichment score. High enrichment scores reflect the expansion of the targets with a given epigenetic mark in the corresponding phylostrata.

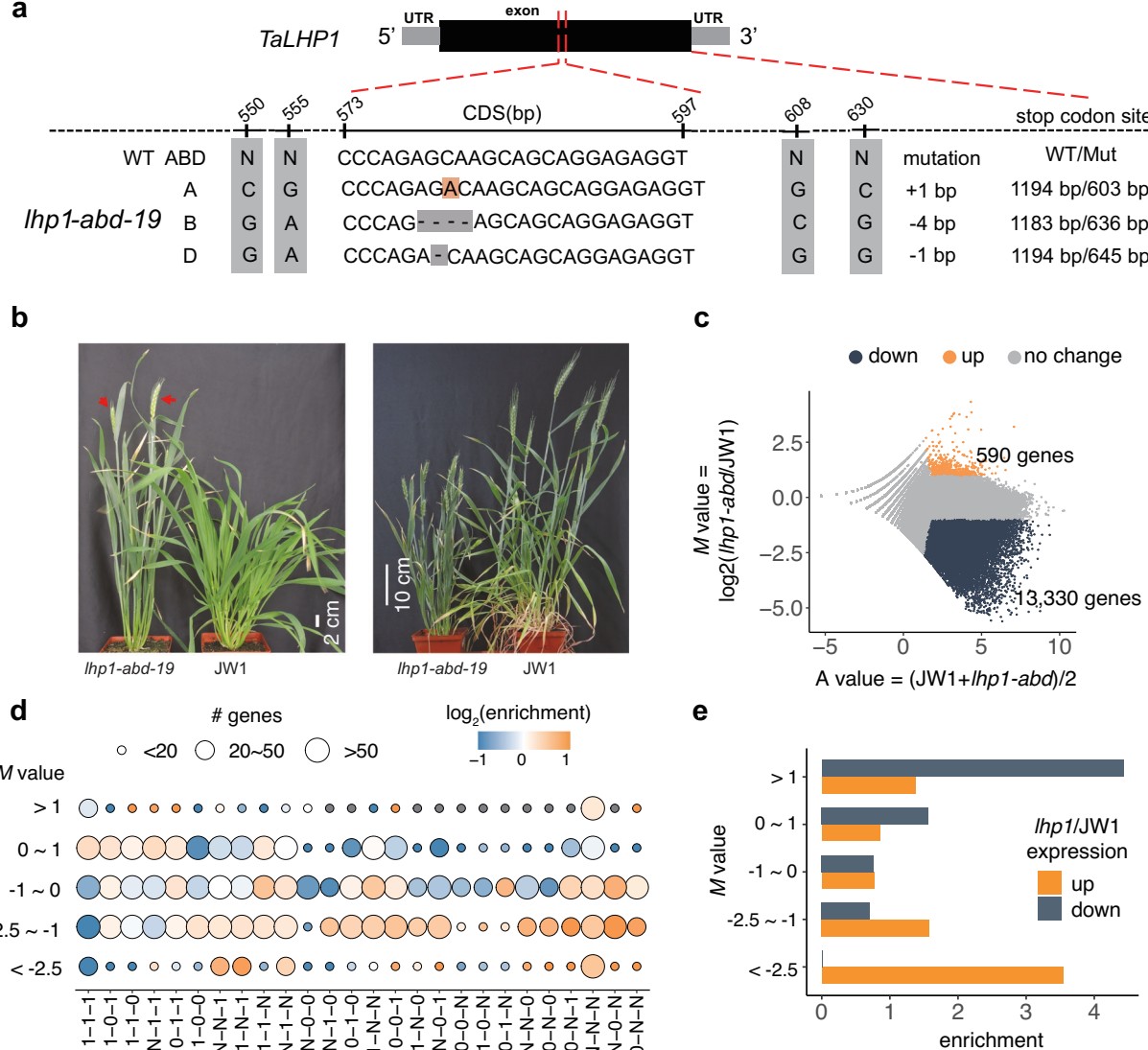

**Fig. 2 | Direct repression of subgenome-diversified genes by LHP1-mediated H3K27me3. a** CRISPR-editing sites in *Talhp1-abd* lines. The positions in gray represent the reference SNVs differentiating homoeologs. The region from 573 to 597 is the CRISPR target site. The type of mutation is listed on the right. **b** Representative developmental phenotypes of the *Talhp1-abd* mutant. Left: heading under long-day conditions, with red arrows indicating spikes; scale bar, 2 cm. Right: growth at 60 days after planting; bar, 10 cm. **c** H3K27me3 changes in the *Talhp1-abd* mutant. The x-axis represents the average read densities in

H3K27me3 target genes in the wild-type and *Talhp1-abd* samples, whereas the y-axis represents the log$_2$-transformed changes (M value) in H3K27me3 in *Talhp1-abd*. **d** Enrichment of homoeologous groups among the genes with differential H3K27me3 changes (M value) in *Talhp1-abd*. Dot colors represent the enrichment score, with all high-confidence genes as the background. **e** Enrichment of genes with up- and downregulated expression among the genes with differential H3K27me3 changes [represented by $M = \log_2$(fold-change)], with all high-confidence genes as the background.

variations and protects the diversified genes from strong purifying selection[46]. The genomic tracks in Fig. 3d reflect the quantitative associations among LHP1 binding, H3K27me3 modifications, and population diversity. In accordance with these findings, population-level investigations of defective epigenetic enzymes in fungi

demonstrated epigenetic repression, typically via H3K27me3, promotes mutations and adaptations[29,30]. Moreover, a recent genome-wide characterization of crossover-active regions and relevant factors in common wheat suggested that H3K27me3 is positively associated with crossovers[48]. These findings and evidence imply that LHP1

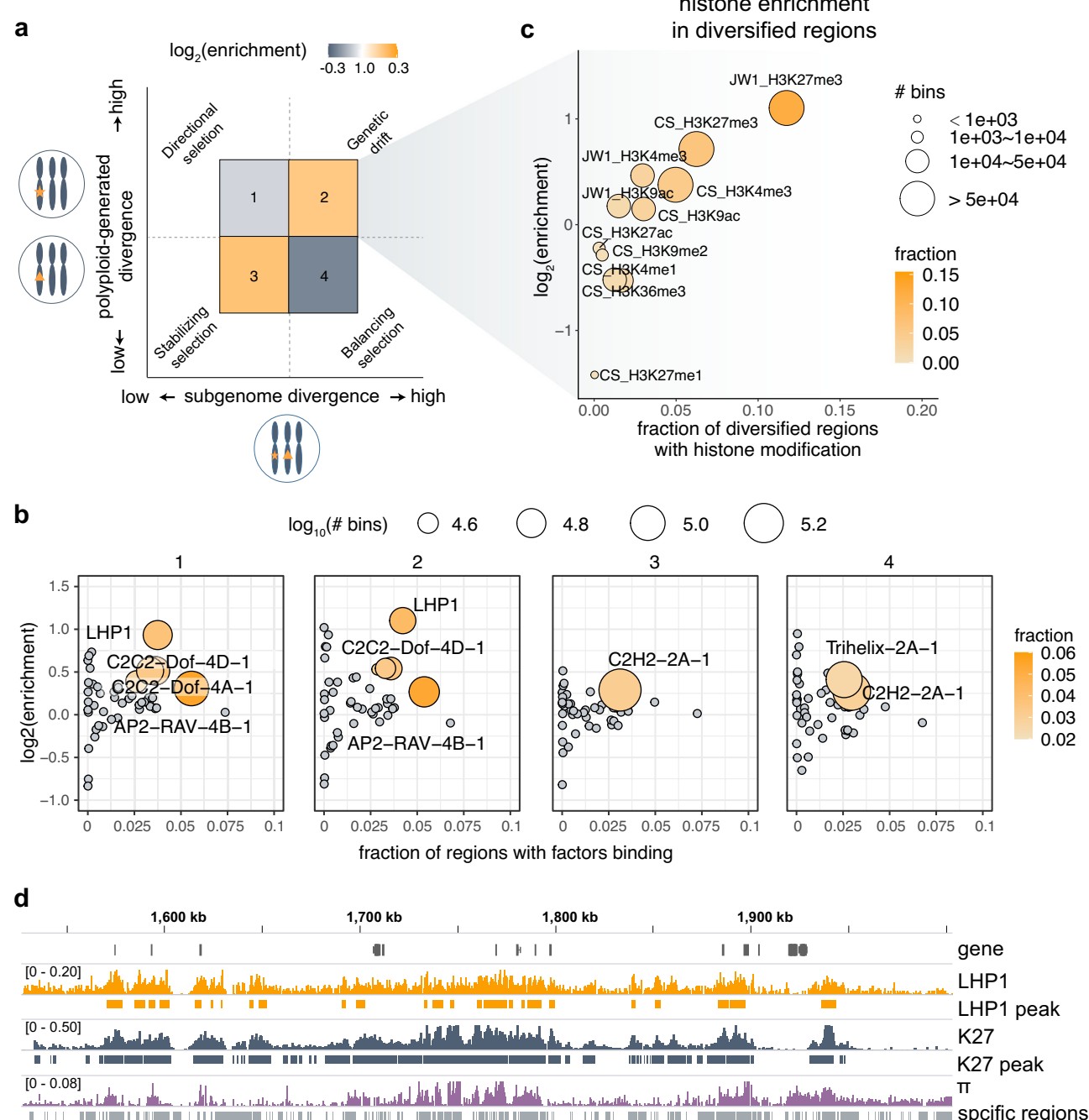

**Fig. 3 | Quantitative association between LHP1 binding and polyploidy-promoted subgenome diversity. a** Association between captured subgenome variations and polyploidy-promoted diversity. The captured variations are reflected by the differences between subgenomes within the same cell. The polyploidy-promoted diversity was evaluated on the basis of population diversity (π). **b** Enrichment of the binding of trans-factors to the four regions shown in panel (**a**).

**c** Enrichment of epigenetic marks in subgenome-specific regions associated with high population diversity. The epigenetic data were generated using Chinese Spring (CS) and JW1. **d** Genomic tracks illustrating the genomic distribution of population diversity (π), subgenome-specific regions, and the loci of LHP1-binding sites and H3K27me3.

repression protects and potentially promotes the accumulation of neutral or weakly deleterious variations across subgenomes.

### Direct repression of subgenome-diversified defense cascades by LHP1

We next examined the biological processes directly influenced by LHP1-mediated H3K27me3. The scatter plots in Fig. 4a present the changes in H3K27me3 and functional gene expression in common wheat. Genes with decreased H3K27me3 levels and increased

expression levels are likely directly repressed by LHP1. Consistent with the phenotype of the *Talhp1-abd* mutant (Fig. 2b), a group of well-characterized genes involved in flowering and/or floral development are targeted by LHP1, including *FUL2* (from subgenomes A, B, and D), *WLHS1* (from subgenome B), *WSOC1* (from subgenome B), and *VRN1* (from subgenome A) (Supplementary Fig. 4)[49,50]. Many LHP1 target genes encode defense-related proteins, including those involved in the perception of pathogens and signaling pathway cascades. In addition to thoroughly characterized defense genes, including the chitin

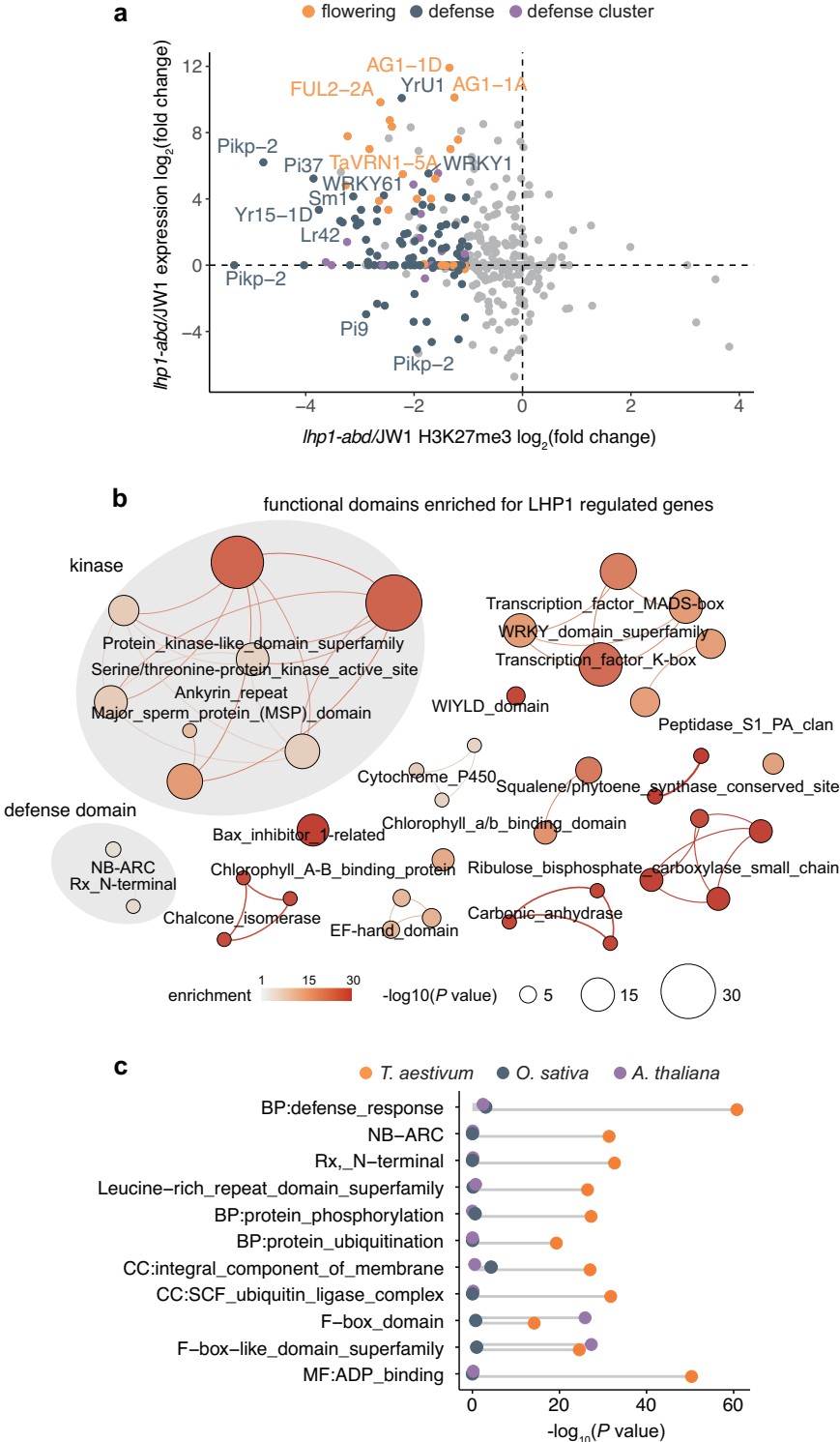

**Fig. 4 | LHP1 preferentially represses defense cascades enhanced in wheat.**
**a** Scatter plot of the changes in the expression and H3K27me3 of functional genes in *Talhp1-abd*. Different colors and shapes represent different functions.
**b** Clustering of functional domains enriched among the genes directly regulated by

LHP1 with decreased H3K27me3 and increased expression in *Talhp1-abd*.
**c** Enrichment of the top-ranked terms assigned to common wheat H3K27me3 target genes in rice and *Arabidopsis*.

sensor-encoding *CERK1*, which is responsive to a wide range of pathogens[51,52], *PR1*, which confers resistance to a broad spectrum of pathogens[53], and stress-responsive WRKY TF genes, the expression of the recently reported four subgenome-diversified gene clusters mediating the synthesis of defense-related metabolites[54] was apparently activated in the *Talhp1-abd* mutant (Supplementary Fig. 4). These

results were consistent with the enriched functions among the LHP1-affected genes (Fig. 4b). The enrichment of H3K27me3 in defense genes was much more apparent in wheat than in rice or *Arabidopsis* (Fig. 4c); most of these genes were from expanded families in Triticeae species (Fig. 1h). The expansion of defense-related gene families in Triticeae is likely associated with increased adaptability. Additionally,

the enrichment of H3K27me3 among these genes represents an epigenetic buffering system that modulates environmental plasticity. These findings imply that LHP1 preferentially represses subgenome-divergent defense pathways mediated by gene families that expanded in Triticeae.

### H3K27me3 repression of subgenome-diversified defense genes is eliminated by stripe rust infection

We next focused on the type of pathogen response preferentially regulated by LHP1. We designed a statistical approach that integrates publicly available transcriptomic data. The underlying principle of this approach is that effectors in the same pathway likely trigger overlapping downstream cascades, which may be reflected by the similarity of the transcriptomic changes. For example, if mutations in two genes trigger similar differential expression patterns, it is likely that these two genes are functionally relevant. We obtained the transcriptomes of samples resistant and susceptible to various pathogens. For each pathogen, the transcriptomic changes were compared between the resistant and susceptible samples. The correlation between the transcriptomic changes revealed by these pairwise comparisons and *Talhp1-abd* mutant-induced changes was assessed (Fig. 5a). The sample with transcriptomic changes that were most enriched with *Talhp1-abd* mutant-induced changes was identified as a near isogenic line (FLW29) containing a locus mediating the resistance to *Puccinia striiformis* f. sp. *tritici* (*Pst*)[55], which causes stripe rust, one of the most widely destructive wheat diseases[53,56]. In other words, the lack of *LHP1* triggered similar transcriptomic changes as the introduction of a *Pst*-resistant locus.

We therefore investigated the effect of *Pst* infection on genome-wide H3K27me3 modifications and the transcriptome. Comparing pre- and post-inoculation H3K27me3 levels in seedlings revealed an overall decrease in H3K27me3, which was highly consistent with H3K27me3 decrease in *Talhp1-abd* mutants (blue dots in Fig. 5b). Only targets with reduced H3K27me3 levels after inoculation and in *Talhp1-abd* mutants were primarily associated with defense responses (Fig. 5c), implying that these defense-associated H3K27me3 loci may be controlled by LHP1. By integrating *Pst*-induced transcriptome changes, we detected elevated transcription of a number of reported defense genes, accompanied by reduced H3K27me3 levels post-inoculation (Fig. 5d). The genomic tracks in Fig. 5e illustrates *Pst* infection-triggered H3K27me3 reduction and expression induction of the homologs of well-characterized defense genes. Thus, H3K27me3 represents an epigenetic buffer system that prevents stochastic processes from altering development by repressing defense genes mostly divergent across subgenomes. Pathogen infection unleashes pre-existing latent subgenomic variations by eliminating buffer activity.

### Knocking out *LHP1* confers resistance to wheat stripe rust

Despite the association between H3K27me3 and defense gene expression in common wheat revealed in this study and reported previously[25,26,48], correlation does not imply causation[57]. We inoculated *Talhp1-abd* wheat plants with *Pst* race CYR32. Compared with the control, *Talhp1-abd* seedlings were highly resistant to *Pst*, with significantly fewer stripe rust sporulation sites and an apparently lower *Pst* DNA-to-host DNA ratio at 14 days post-infection (dpi) (Fig. 6a). The fungus was visualized by staining leaf segments with wheat germ agglutinin at 2 dpi. The average hyphal area was significantly smaller (less than half) for the *Talhp1-abd* plants than for the JW1 plants (Fig. 6b). Reactive oxygen species (ROS) are key signaling molecules that enable cells to rapidly respond to external stimuli[58]. We quantified the ROS content at 2 dpi via 3,3′-diaminobenzidine (DAB) staining, which showed that the $H_2O_2$ level was higher in the *Talhp1-abd* mutant than in the control (Fig. 6c). These results indicate that the CRISPR–Cas9 editing of *LHP1* leads to stripe rust resistance associated with ROS-stimulated defenses, highlighting the regulatory role of LHP1 on the response of polyploid wheat to environmental stresses.

Considered together, the results of the present study revealed the repressive effects of LHP1 on the expression of subgenome-diversified defense genes, which protect developmental processes from stochastic external changes and potentially promote neutral drift and diversity. Pathogen infections adversely affect the buffering system, enabling previously unavailable phenotypic variants to surface, thereby releasing defense cascades and facilitating the timely fixation of favorable mutations in particular environmental niches (Fig. 7). The effects of the epigenetic buffering and release of subgenome diversity lead to developmental robustness and the timely manifestation of latent phenotypes, which helps to explain the selective advantage and phenotypic plasticity of polyploid wheat.

## Discussion

In this study, we clarified how the epigenetic buffering system coordinates internal genetic changes in response to external stimuli in polyploid wheat. Accordingly, we addressed the long-standing question of how genetic diversity due to polyploidization is interpreted and regulated to mediate environmental plasticity. These findings clarified the LHP1 function in terms of evolution, development, and practical applications.

### LHP1 is the epigenetic anchor linking subgenome diversity and adaptive evolution in polyploids

During long-term evolution, epigenetic buffering systems may help populations reach a local fitness optimum, while also promoting neutral accumulation of potentially selectable polymorphisms in the absence of internal or external interventions. External stresses reveal cryptic phenotypes by releasing genetic variations, which are converted to a non-neutral state, ensuring that deleterious variations are periodically purged from the population and favorable variations are fixed. Because rare combinations of variations may produce a new and advantageous phenotype, increases in the diversity of allopolyploids increase the likelihood that plants will manifest advantageous traits pushing the population upward on the adaptive landscape. This represents a molecular mechanism by which adaptive peak shifts occur without adaptive valleys. Therefore, the epigenetic buffering system coupled with environmental stimuli likely shaped the trajectory of subgenome diversification and adaptive evolution in polyploid wheat.

The enrichment of H3K27me3 in subgenome-diversified genes was also observed in other polyploids, including cotton, *Brassica* species, and *Arabidopsis*, but with relatively weaker signal than those in hexaploid and tetraploid wheat species (Supplementary Fig. 5). Thus, the PcG-mediated epigenetic control of subgenome diversity is likely a common mechanism among polyploids.

### LHP1 ensures appropriate developmental responses to external stimuli

The buffering effect of LHP1 on subgenomic diversity enables common wheat to resist the effects of stochastic processes and develop normally, while also contributing to defense responses to pathogen infections, ultimately resulting in environmental plasticity. These seemingly diverse effects of LHP1 are readily encompassed by a simple epigenetic framework, which presents a rich subject for further analysis.

Polyploidy is a hallmark of cancer[59]. The high plasticity of cancer cells in tumors has been attributed to high genetic heterogeneity[59], which is similar to the close association between polyploid plasticity and subgenomic diversity. Polycomb group proteins have been implicated in cancer development and progression. Their catalytic subunit recurrently mutates in several forms of cancer and is highly

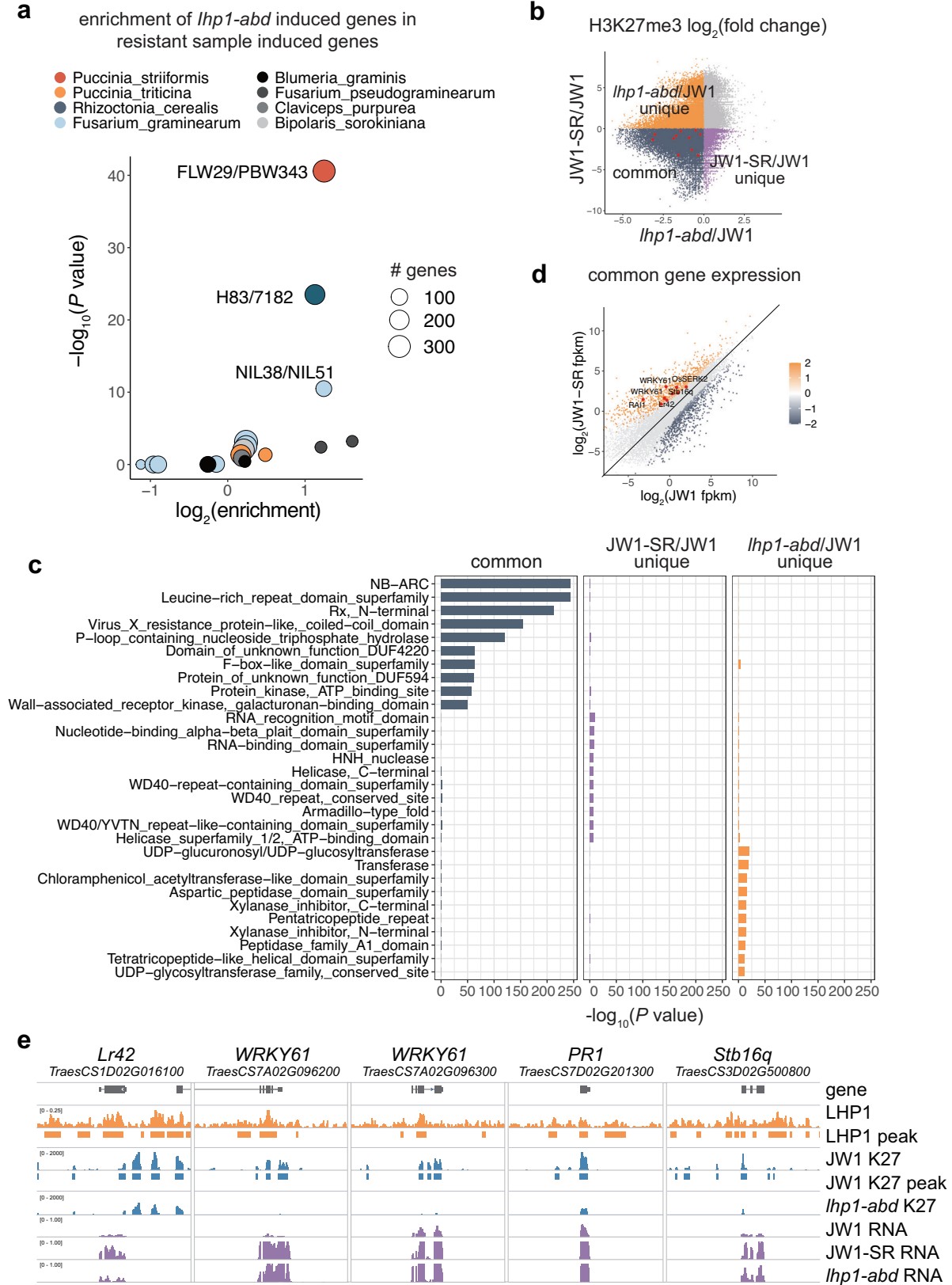

**Fig. 5 | Stripe rust infection induces subgenome-diversified defense gene expression by removing H3K27me3. a** Enrichment of *Talhp1-abd*-induced genes among the genes more highly expressed in samples resistant to various pathogens (represented by different colors) than in the corresponding susceptible samples. **b** Scatter plot representing H3K27me3 changes post-inoculation and in the *Talhp1-abd* mutant. Reported defense genes are marked in red and their expression is characterized and labeled in panel (**d**). **c** Enriched functional domains among the genes with H3K27me3 levels commonly or uniquely decreased in *Talhp1-abd*. **d** Stripe rust-induced changes in the expression of genes. Reported defense genes marked in panel (**b**) are labeled. **e** Genomic tracks illustrating subgenome-diversified defense gene expression and H3K27me3 changes in *Talhp1-abd*.

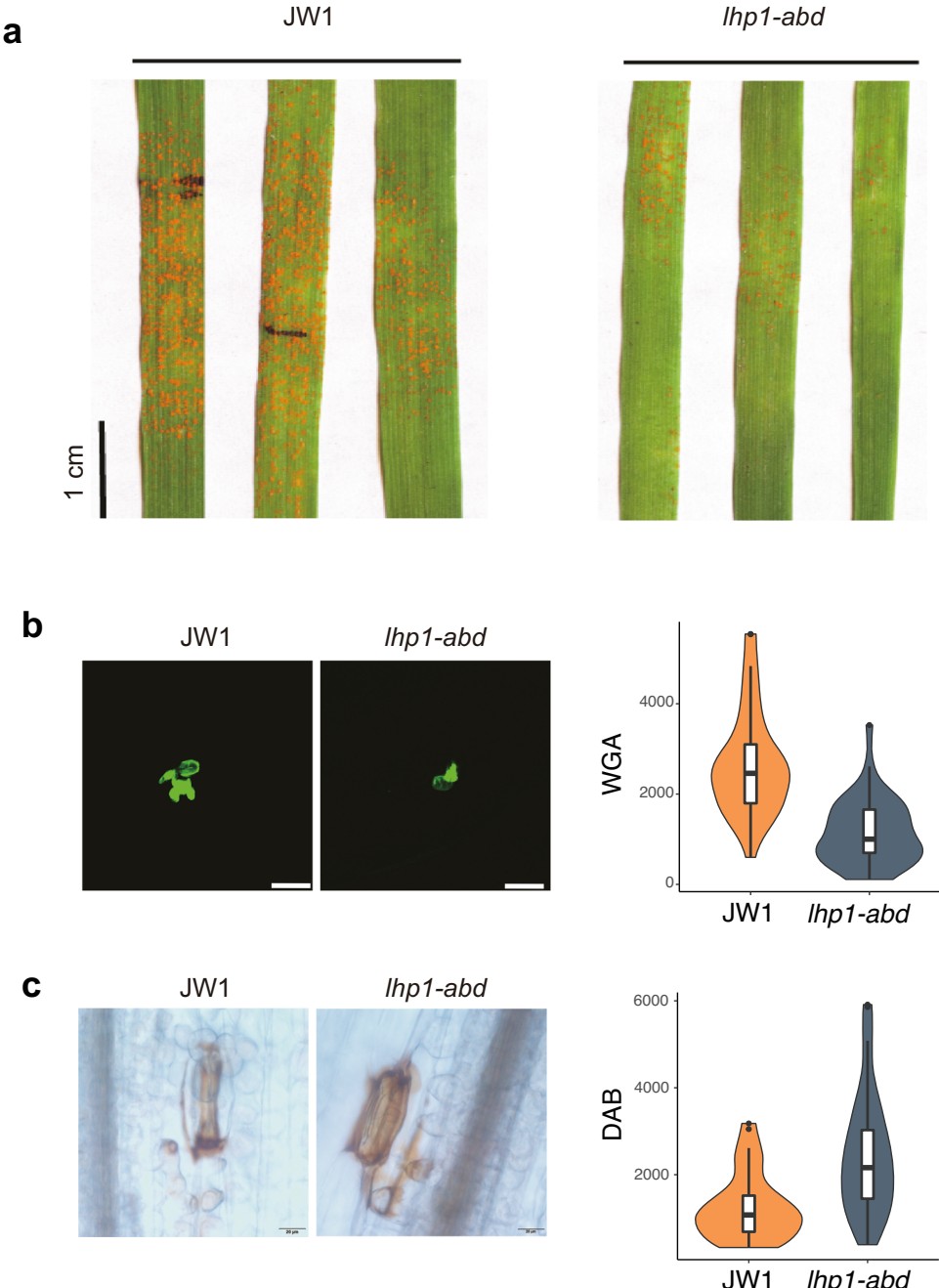

**Fig. 6 | CRISPR–Cas9-mediated inactivation of three homoeoalleles of *LHP1* confers resistance to stripe rust. a** *Pst* growth in wild-type and *Talhp1-abd* plants at 2 dpi (stained with wheat germ agglutinin conjugated to Alexa-488); bar, 1 cm. **b** Left: wild-type and *Talhp1-abd* plants inoculated with *Pst* CYR32 and photographed at 14 dpi; bar, 50 μm. Right: quantification of *Pst* growth in infected JW1 and *Talhp1-abd* leaves at 2 dpi by wheat germ agglutinin staining. Values represent the mean ± SD (error bars) (*n* = 40). For JW1, mean value = 1563, median = 1468, SEM = 100, SD = 629; For *Talhp1-abd*, mean value = 1227, median = 1101, SEM = 75, SD = 472. Horizontal lines in boxplots show median, hinges show IQR, whiskers show 1.5 × IQR, points beyond 1.5 × IQR past hinge are shown. **c** Left: $H_2O_2$ accumulation in wheat leaves at 2 dpi revealed by DAB staining; bar, 20 μm. Right: quantification of $H_2O_2$ accumulation in infected JW1 and *Talhp1-abd* leaves at 2 dpi by DAB staining (*n* = 40). For JW1, mean value = 1999, median = 1683, SEM = 178, SD = 1128; For *Talhp1-abd*, mean value = 2945, median = 2696, SEM = 234, SD = 1481. Boxplots definition is the same as (**b**).

expressed in numerous other types of cancer[60,61]. The oncogenic role of PcGs is under increased scrutiny because of the potential utility of these proteins for developing cancer therapeutics[60,61]. However, the underlying mechanism remains unclear and may primarily involve transcriptional repression[61]. It will be interesting to see whether our findings regarding the translation of genetic diversity to environmental plasticity and adaptive evolution via the buffering effect of PcGs are applicable to cancer cells.

**Deliberate manipulation of LHP1-mediated epigenetic buffering potentially releases latent genetic diversity**
The findings presented herein provide intriguing insights relevant to exploiting intrinsic immunity to enhance defenses. Identifying resistance genes or loci in wild relatives and introducing them into modern cultivars has been the main strategy used to improve common wheat defense responses. It is thought that common wheat lacks sufficient diversity to cope with environmental stresses because of polyploidy-related

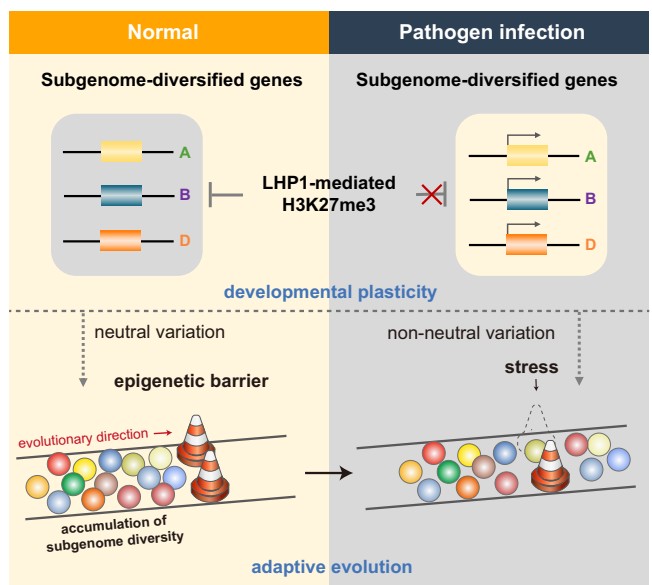

**Fig. 7 | Model illustrating the LHP1-mediated conditional repression of subgenome-diversified defenses, which leads to developmental plasticity and facilitates adaptive evolution in allopolyploid wheat.** The expression of subgenome-diversified genes is repressed by LHP1-mediated H3K27me3 under normal conditions, which protects normal development from stochastic environmental changes and promotes neutral or nearly neutral variations. Under stress conditions, the epigenetic repression is eliminated and the hidden subgenome diversification is manifested, thereby facilitating the timely fixation of favorable changes in different evolutionary niches. The extensive diversification of allopolyploid wheat increases the likelihood the population will be pushed upward on the adaptive landscape. Thus, LHP1 helps protect and potentially promote the accumulation of subgenomic diversity, while also ensuring appropriate responses to environmental stimuli, leading to developmental plasticity and facilitating adaptive evolution in polyploid wheat.

bottlenecks[31]. The genomes of wheat and other crops contain a large proportion of inactive or silent genetic loci, many of which are related to key agronomic traits and stress resistance and would considerably increase genetic diversity if properly modulated[62]. In the current study, we demonstrated that although the common wheat cultivar 'JW1' is highly susceptible to stripe rust, modifying an epigenetic factor is sufficient for conferring resistance to stripe rust. The customized removal of gatekeepers provides a strategy for activating otherwise silent intrinsic defense responses. In addition, manipulating the epigenetic buffering system may facilitate the detection of key defense-related genes that are currently hidden in the wheat genome.

The introduction of an external superior allele does not necessarily result in robust superior phenotypes in all environments. Our data suggest that LHP1 is an epigenetic gatekeeper responsive to environmental changes and may ultimately determine the successful phenotypic expression of latent genetic diversity. Further characterizing the epigenetic buffering specificity may promote the adaptation of superior alleles in different hosts and various ecological niches.

Taken together, the findings of this study imply that the deliberate manipulation of the epigenetic-based storage, buffering, and occasional release of genetic variations will facilitate the detection and exploitation of useful silent loci in modern cultivars and promote the breeding of improved crops and food sustainability.

## Methods
### Plant materials and growth conditions
Common wheat (*Triticum aestivum*) variety 'JW1' seeds were surface-sterilized via a 10-min incubation in 30% $H_2O_2$ and then thoroughly washed five times with distilled water. The seeds were germinated in water for 3 days at 22 °C, after which the germinated seeds with residual endosperm were transferred to soil. The seedlings were cultivated under long-day conditions.

### Constructs for gene editing and wheat transformation
To modify all three *TaLHP1* copies, we used two sgRNAs that target conserved regions (target1: AGGTCCTATGGCAAGCGCAA, target2: GAGCAAGCAGCAGGAGAGGT). The sgRNAs were identified by CasOT[63]. Synthesized oligos for target-specific sgRNAs were annealed and cloned into the pBUE411 vector.

### *Agrobacterium tumefaciens*-mediated common wheat transformation
After sequencing the target sites, the binary vector was transformed into the wheat cultivar JW1 by *Agrobacterium tumefaciens*-mediated transformation[64,65]. Briefly, wheat spikes were collected at anthesis, harvested 14–16 days postanthesis (DPA) and sterilized with 75% ethanol for 30 s, then with 1% sodium hypochlorite (NaClO) for 15 min, and finally rinsed 5 times with sterile water under aseptic conditions. Immature embryos were isolated and incubated with *Agrobacterium* strain EHA 105 for 5 min. After cocultivation at 25 °C for 2 days in darkness, the embryonic axes were removed with a scalpel, and the scutella were transferred onto plates. After 5 days, the tissues were then transferred to callus selection medium for 2 weeks. The immature embryos were then placed on an induction medium for 3 weeks. The calli were then differentiated under continuous illumination (5000 lx) with fluorescent lights at 25 °C for 14 days. The regenerated shoots were transferred to the root elongation medium. The rooted plantlets were then transferred into pots and grown in growth chambers, where they were cultivated at a temperature of 20 °C with a light intensity stronger than 60,000 lx and a night temperature of 16 °C. Transgenic wheat plants were generated by Professor Genying Li (Shandong Academy of Agricultural Sciences, China).

### Mutant screening
To determine the editing efficiency, we amplified the two target sites in $T_0$ transgenic plants for Sanger sequencing and found that the first target site was not edited, and the second target site was edited in all three subgenomes. Therefore, in the subsequent genotyping, we used Hi-TOM[66] (http://hi-tom.net/hi-tom/) to detect only the second target sites. We amplified the second target site with site-specific primers (forward: GGAGTGAGTACGGTGTGCCGTTCAGATCCTCGGTCTCT, reverse: gagttggatgctggatggATTTGAGCTGCCCTCCTGTGT, tsingke), then the products of the first-round PCR were sent to Hi-TOM sequencing. We selected two homozygous mutants (*lhp1-abd-18* and *lhp1-abd-19*) that resulted in premature termination of the LHP1 protein for further research. The results in the main text are obtained using *lhp1-abd-19*. All major conclusions are consistent using the other mutant line (Supplementary Figs. 6–9). All data were generated in biological duplicates (Supplementary Data 2).

### *Puccinia striiformis* f. sp. *tritici* inoculations
We collected fresh urediospores of *Pst* race CYR32 from the leaves of wheat cultivar MingXian169. We dispersed the fresh spores in tap water and spread equal contents on the second leaf of control and transgenic wheat plants. The infected plants were incubated in a humid chamber overnight at 16 °C in the dark. We transferred the infected plants to a growth chamber (16 °C at 16 h light/8 h dark, 80% RH) to allow the growth of rust pathogens. At 48 h post-inoculation (hpi), we stained the leaf samples with 3′,3′-diaminobenzidine (DAB) (Merck Sigma–Aldrich, Shanghai, China) to examine the accumulation of $H_2O_2$. Only the site where an appressorium had formed over a stoma was considered a successful penetration. We stained the *Pst* hyphae with wheat germ agglutinin conjugated to Alexa-488 (Invitrogen, Waltham, MA, USA)[67]. At 14 dpi, we scanned the *Pst*-infected leaves to examine

the phenotypes and count the *Pst* uredial pustule number. We then extracted genomic DNA and quantified the fungal biomass by quantitative PCR (with primers PstEF1-F: TTCGCCGTCCGTGATATGAGA-CAA; PstEF1-R: ATGCGTATCATGGTGGTGGAGTGA; TaEF-S: TGG TGTCATCAAGCCTGGTATGGT; TaEF-AS: ACTCATGGTGCATCTCAA CGGACT)[68].

## ChIP and RNA sample preparation

At the two-leaf stage, seedling leaves of 14-day *Pst*-inoculated and control were either frozen in liquid nitrogen for an RNA isolation step or vacuum-infiltrated with a formaldehyde cross-linking solution (0.4 M sucrose, 10 mM Tris [pH 8], 1 mM EDTA, 1% formaldehyde) for the ChIP assay.

## ChIP-seq assay

Crosslinked materials were ground into fine powder with liquid nitrogen, resuspended in ChIP Lysis Buffer 1 (CLB1: 50 mM HEPES [pH 7.5], 150 mM NaCl, 1 mM EDTA, 1% Triton X-100, 10% glycerol, 1x inhibitor cocktail, 0.035% 2-mercaptoethanol) and incubated for 60 min with rotation at 4 °C. After incubation, the nucleus was collected after filtering the mixture through a 40-μm strainer, centrifuging at $3000 \times g$ for 30 min at 4 °C in a swinging bucket rotor and removing the supernatant. The nucleus was washed twice with ChIP Lysis Buffer 2 (CLB2: 50 mM HEPES [pH 7.5], 150 mM NaCl, 1 mM EDTA, 1% Triton X-100, 10% glycerol, 1x inhibitor cocktail). DNA was sheared by sonication to approximately 300- to 500-bp fragments. After centrifugation (10 min at 13,000 rpm), the supernatant was precleared with 40 μl salmon sperm (SS) DNA/Protein A agarose for 60 min at 4 °C. After 2 min of centrifugation at $500 \times g$, the supernatant was transferred to a siliconized tube, and 10 μl of the appropriate antibody (H3 trimethyl-Lys 27 (ABclonal, A2363), H3 trimethyl-Lys 4 (Abcam, Cambridge, England), and H3 acetyl-Lys 9 (Millipore)) was added. After incubation overnight with rotation, 40 μl SS DNA/Protein A agarose was added, and incubation continued for 1 h. The agarose beads were then washed with 1 ml of each of the following: Low salt buffer (50 mM HEPES [pH 7.5], 150 mM NaCl, 1 mM EDTA), High salt buffer (50 mM HEPES [pH 7.5], 635 mM NaCl, 1 mM EDTA), LiCl wash buffer (0.25 M LiCl, 0.5% NP-40, Tris-HCl [pH 8], 1 mM EDTA), and 1× TE (10 mM Tris-HCl [pH 8], 1 mM EDTA). The immunocomplexes were eluted from the beads with 400 μl 1% SDS, 0.1 M NaHCO₃. A total of 20 μl of 5 M NaCl was then added to each tube, and crosslinks were reversed by incubation at 65 °C for 5–6 h. Residual protein was degraded by the addition of 20 μg Prot K (in 10 mM EDTA and 40 mM Tris [pH 8.0]) at 45 °C for 1 h, followed by phenol/chloroform/isoamyl alcohol extraction and ethanol precipitation. Pellets were washed with 70% EtOH and resuspended in 30 μl TE buffer. More than 10 ng ChIP DNA was used to prepare each sequencing sample. Libraries were constructed and sequenced by Novogene (Beijing, China). The libraries were sequenced with the Illumina NovaSeq 6000 system to produce 150-bp paired-end reads.

For spike-in ChIP, chromatin from *Arabidopsis* was used as a spike-in control. Briefly, fragmented chromatin from all samples (including wheat and *Arabidopsis*) was isolated according to the normal ChIP protocol. Then, 40 μl of chromatin from all samples was reverse-crosslinked, digested with proteinase K, and subjected to DNA purification. The concentration of purified DNA was measured with Qubit dsDNA High-Sensitivity Assays (Invitrogen). Depending on the DNA concentration of each sample, chromatin from *Arabidopsis* and wheat are mixed together in a ratio of 1:100 for immunoprecipitation. The rest of the procedures are the same as the normal ChIP.

## LASSO regression analysis

To determine which of the 191 factors would be useful for predicting subgenome-diversified genes, we used the R package glmnet[69] to perform the least absolute shrinkage and selection operator (LASSO) regression model analysis for feature selection:

$$\log(Y_i) = \sum_{j=1}^{M} \beta_j A_{ij} + \gamma \qquad (1)$$

where $Y_i$ is the diversity for gene i, $A_{ij}$ is the jth factor that targets the ith gene and $\beta_j$ is the fitted coefficient for the jth factor. M is the total number of factors. $\gamma$ is the intercept term calculated by the model. The goal of the algorithm is to minimize:

$$\sum_{i=1}^{N} \left( y_i - \sum_{j=1}^{M} A_{ij}\beta_j \right)^2 + \lambda \sum_{j=1}^{M} |\beta_j| \qquad (2)$$

where $\lambda$ controls the strength of the L1 penalty. $N$ is the total number of.

LASSO analysis combined with 10-fold cross-validations was conducted to choose the most useful predictive features, and the model family we used was "binomial".

## LHP1 ChIP-seq assay in wheat protoplasts

ChIP assays using JW1 leaf protoplasts were performed. JW1 plants were grown on soil under 16 h light/8 h dark conditions for 2 weeks before protoplast isolation. Approximately 30 μg of pMD19-T plasmids containing *p35S*:3flag-TaLHP1-B DNA were transfected into leaf protoplasts using the PEG-mediated transfection method. After incubating the protoplasts at room temperature for 48 h under dark conditions, the protoplasts were crosslinked with 1% formaldehyde in W5 solution for 10 min on ice and quenched with 32 μl 2 M glycine for 5 min. Protoplasts were collected by centrifuging at $600 \times g$ for 2 min at 4 °C, washed with 500 μl W5 solution once and collected again. Protoplasts were lysed in 120 μl of room temperature lysis buffer (50 mM Tris-HCl pH 8.0, 10 mM EDTA, 1% (wt/vol) SDS, 1 mM PMSF, 1X protease inhibitor cocktail) by vortexing. Total lysates containing chromatin were subjected to sonication by Bioruptor until the chromatin was fragmented into 300- to 500-bp fragments. Another 400 μl RIPA ChIP buffer (10 mM Tris-HCl pH 7.5, 140 mM NaCl, 1 mM EDTA, 0.5 mM EGTA 1% (vol/vol) Triton X-100, 0.1% (wt/vol) SDS, 0.1% (wt/vol) Na deoxycholate, 1 mM PMSF, 1X protease inhibitor cocktail) was added to the lysates. The lysates were centrifuged at $12,000 \times g$ for 10 min at 4 °C, and the supernatant was transferred to a new tube. Another 410 μl of RIPA ChIP buffer was mixed with the remaining pellet, and centrifugation, as described above, was performed again to obtain the second supernatant. The two rounds of supernatant were pooled, and the volume was brought to 1 ml with RIPA ChIP buffer. Keep 100 μl of chromatin as 10% input. Then, 20 μl agarose beads conjugated with anti-Flag antibody (A2220, Sigma) were added to the chromatin suspension and incubated for 2 h at 4 °C. After binding with chromatin, the beads were subsequently washed twice with RIPA buffer (10 mM Tris-HCl pH 7.5, 140 mM NaCl, 1 mM EDTA, 0.5 mM EGTA 1% (vol/vol) Triton X-100, 0.1% (wt/vol) SDS), LiCl buffer (0.25 M LiCl, 1% (wt/vol) Na deoxycholate, 10 mM Tris-HCl pH 8.0, 1% NP-40, 1 mM EDTA) once, and TE (10 mM Tris-HCl pH 8.0, 10 mM EDTA) buffer once. The protein–DNA complexes were eluted from beads by adding 150 μl of complete elution buffer (20 mM Tris-HCl pH 7.5, 5 mM EDTA, 50 mM NaCl, 1% (wt/vol) SDS, 50 mg/ml proteinase K) for 2 h at 68 °C with agitation at 1300 rpm. The eluate was then transferred to a new tube. The beads were eluted again with 150 μl of elution buffer (20 mM Tris-HCl pH 7.5, 5 mM EDTA, 50 mM NaCl) for 5 min. The two rounds of eluates were combined. During the elution step, the input DNA was prepared by adding 200 μl elution buffer and 7.5 μl proteinase K (20 mg/ml) and incubating at 68 °C for 2 h. ChIP DNA was extracted with phenol:chloroform (1:1), precipitated with ethanol and resuspended in TE buffer to prepare the ChIP-seq library using the

ThruPLEX DNA-seq Kit. The libraries were sequenced with Hiseq-PE150 to produce 150 bp paired-end reads by Novogene (Beijing, China).

## Processing of ChIP-Seq and RNA-seq data

Sequencing reads were cleaned with fastp (version 0.20.0)[70], which eliminated bases with low-quality scores (<25) and irregular GC contents, sequencing adapters, and short reads. The remaining cleaned reads were mapped to the International Wheat Genome Sequencing Consortium (IWGSC) reference sequence (version 1.0) with the Burrows–Wheeler Aligner (version 0.7.17-r1188)[71] for wheat ChIP sequencing. For spike-in ChIP-seq, cleaned reads were mapped to the merged genome of common wheat (IWGSC) and *Arabidopsis thaliana* (TAIR10). The HISAT2 program (version 2.2.1)[72] was used for mapping the RNA sequencing (RNA-seq) reads to the reference sequences.

The MACS (version 2.2.6)[73] program was used to identify the read-enriched regions (peaks) of the ChIP-Seq data with the cutoff $P < 1e-10$. To quantify gene expression levels, the featureCount program of the Subread package (version 2.0.0)[74] was used to determine the RNA-seq read density for the genes. To compare expression levels across samples and genes, the RNA-seq read density of each gene was normalized based on the exon length in the gene and the sequencing depth (i.e., fragments per kilobase of exon model per million mapped reads). To quantify histone markers across genes for the figure prepared with Integrative Genomics Viewer[75], the number of reads at each position was normalized against the total number of reads (reads per million mapped reads). For spike-in ChIP-seq, the number of reads at each position was normalized against the total number of reads mapped to *A. thaliana* (reference-adjusted reads per million, RRPM). The edgeR program[76] was used to detect differentially expressed genes based on the combined criteria: |$\log_2$ fold-change| > 1 and FDR (Benjamini–Hochberg corrected) <0.05. The MAnorm2 package[77] was used for the quantitative comparison of ChIP-Seq signals between samples with the following criteria: |M value| > 1 and $P < 0.05$. For spike-in ChIP-seq, the value of RRPM was used to compare signals between samples with the following criteria: |$\log_2$ fold-change| > 1 and RRPM > 5.

## Identification of sc-triads and non-sc-homoeologs

Orthofinder[78] was used to identify the homologous genes between *A. thaliana*, *Zea mays*, *Oryza sativa*, *Brachypodium distachyon*, *Hordeum vulgare*, *Secale cereale*, *Triticum urartu*, and *Aegilops tauschii*. *Triticum turgidum*, *T. aestivum*. Each subgenome was treated as an individual genome. The orthogroups with only one copy in each subgenome (1:1:1) were defined as sc-triads, and others were defined as non-sc-homoeologs. The GO terms curated by GOMAP[79] and protein domains of Interpro were used to detect the over-represented functional terms and domains associated with the sc-triads and non-sc-homoeologs. Enriched GO terms and Interpro domains were visualized using Gephi (version 0.9.2)[80].

## Phylostratigraphic analysis

Genes were divided into 9 groups by using the genomic phylostratigraphic approach[44]. According to the evolutionary relationship of the above species, genes of *T. aestivum* with homologous relationships to *A. thaliana* were identified as phylostrata 1 (PS1). Genes of *T. aestivum* with homologous relationships to *Z. mays* and absent in *A. thaliana* were identified as PS2. Genes of *T. aestivum* with homologous relationships to *O. sativa* and absent in *A. thaliana* and *Z. mays* were identified as PS3. The genes of PS4 to PS9 were categorized in the same way.

## Detection of subgenome-homologous and subgenome-specific regions

To determine the homologous regions across subgenomes, we used the subgenome alignment results generated by NUCmer. The reciprocal aligned regions that were longer than 400 bp were defined as homologous regions across three subgenomes (homo3) or two subgenomes (homo2). Regions that were not aligned to another two subgenomes were defined as specific regions (specific). Subgenome-syntenic regions were detected using MCScanX (python version)[81], with homologous regions localized to syntenic regions defined as homoeo3, i.e., syntenic homo3 regions. Accordingly, 35%, 15%, 51%, and 16% of the genomic regions were defined as specific, homo2, homo3, and homoeo3, respectively.

## Nucleotide diversity calculation

Nucleotide diversity (π) was calculated from a 100 bp sliding window and a 100 bp step using VCFtools (v0.1.13)[82] based on a total of 100 representative whole-genome resequenced hexaploid wheat accessions[83].

## Statistics and reproducibility

We have replicates for all data generated from wild type, including RNA-seq before and after pathogen treatment, H3K27me3 ChIP-seq before and after pathogen inoculation, H3K27me3 ChIP-seq in leaf protoplast, and ChIP-seq of LHP1-binding loci in leaf protoplast (Supplementary Data 2). We have 2 independent mutant lines. The main results were reproduced in this study (Supplementary Figs. 6–9). No statistical method was used to predetermine the sample size. No data were excluded from the analyses; the experiments were not randomized; the investigators were not blinded to allocation during experiments and outcome assessment.

## Reporting summary

Further information on research design is available in the Nature Portfolio Reporting Summary linked to this article.

## Data availability

The ChIP-Seq and RNA-seq data generated in this study have been submitted to the NCBI Gene Expression Omnibus (GEO; https://www.ncbi.nlm.nih.gov/geo/) under accession number GSE218538 (https://www.ncbi.nlm.nih.gov/geo/query/acc.cgi?acc=GSE218538). Tracks for all sequencing data can be visualized through our local genome browser (http://bioinfo.sibs.ac.cn/LHP1_jbrowse/). The histone ChIP-seq data of Chinese Spring (CS) seedlings used in this study are under accession numbers GSE139019 and GSE121903 in the NCBI GEO database[25,26]. The H3K27me3 ChIP-seq data for *Oryza sativa* and *Arabidopsis thaliana* used in this study were downloaded from the NCBI GEO database (accession numbers GSE67322 and GSE142462)[38,84]. RNA-seq used in Fig. 5a were obtained from NCBI SRA database (accession numbers PRJEB12358, PRJEB24686, PRJNA263755, PRJNA289545, PRJNA401295, PRJNA428316, PRJNA450087, PRJNA595999, PRJNA613349, PRJNA630776, PRJNA664832, PRJNA718488, PRJNA749387). The functional genes of *T. aestivum* were downloaded from WheatOmics (http://wheatomics.sdau.edu.cn/genes/)[85]. *B. distachyon* genomes were obtained from Phytozome (v12)[86]. *O. sativa* from RAP-DB[87]. *Z.mays* from MaizeGDB[88]. *A. thaliana* from TAIR[89]. *H. vulgare* and *T. turgidum* from the Plant Genomics and Phenomics Research Data Repository[90]. *S. cereale* was obtained from the Chinese National Genomics Data Center[91]. *T. urartu* from MBKBase[92]. *A. tauschii* from the EnsemblPlants database (Aet_v4.0)[93].

## Code availability

Scripts are available at https://github.com/yuyun-zhang/hexa_LHP1.

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

## Acknowledgements

This study was supported by State Key Laboratory of Crop Gene Exploration and Utilization in Southwest (SKL-KF202305), National Science Fund for Excellent Young Scholars (32022012) (Y.J.Z.), the National Natural Science Foundation of China (32270628) (Y.J.Z.) and (31972350, 32372557) (J.Y.G.), the European Research Council ERC (Project 101044399-3Dwheat) (M.B.), the Saclay Plant Sciences-SPS (ANR-17-EUR-0007) (M.B.) and the State Key Laboratory of Genetic Engineering (SKLGE-2312) (W.L.Z.). We thank Dr. Fei Lu from the Institute of Genetics and Developmental Biology, CAS, and Dr. Weilong Guo and Dr. Zihao Wang from China Agricultural University for help with population data analysis.

## Author contributions

Y.J.Z., J.Y.G., Z.C.D., M.B. conceived and designed the experiments. L.G.Y., W.L.Z., Y.P.T., Z.J.L., C.H.D., Y.C., S.B.Y., R.Z.Z., H.S.D., L.H.Y., Y.L.Z., W.T., L.H. and G.Y.L. performed the experiments. Y.Y.Z., H.Y.W., M.Y.W., J.Y.L.,Y.L.X., T.F.T., and Y.J.Z. analyzed the data. Y.J.Z. wrote the manuscript with input from all authors.

## Competing interests

The authors declare no competing interests.

## Additional information

[1]State Key Laboratory of Genetic Engineering, Collaborative Innovation Center of Genetics and Development, Department of Biochemistry, Institute of Plant Biology, School of Life Sciences, Fudan University, 200438 Shanghai, China. [2]National Key Laboratory of Plant Molecular Genetics, CAS Center for Excellence in Molecular Plant Sciences, Shanghai Institute of Plant Physiology and Ecology, Shanghai Institutes for Biological Sciences, Chinese Academy of Sciences, 300 Fenglin Road, 200032 Shanghai, China. [3]University of the Chinese Academy of Sciences, 100049 Beijing, China. [4]Frontiers Science Center for Molecular Design Breeding, China Agricultural University, Beijing, China. [5]Guangdong Provincial Key Laboratory of Plant Adaptation and Molecular Design, Guangzhou Key Laboratory of Crop Gene Editing, Innovative Center of Molecular Genetics and Evolution, School of Life Sciences, Guangzhou University, 510006 Guangzhou, China. [6]Henan University, School of Life Science, 457000 Kaifeng, Henan, China. [7]Crop Research Institute, Shandong Academy of Agricultural Sciences, Jinan, China. [8]Ministry of Agriculture, Key Laboratory of Wheat Biology and Genetic Improvement on North Yellow and Huai River Valley, Jinan, China. [9]National Engineering Research Center for Wheat and Maize, Jinan, Shandong, China. [10]The State Key Laboratory of Plant Cell and Chromosome Engineering, Institute of Genetics and Developmental Biology, the Innovative Academy of Seed Design, Chinese Academy of Sciences, 100101 Beijing, China. [11]Northwest Institute of Plateau Biology, Chinese Academy of Sciences, 810008 Xining, China. [12]State Key Laboratory of Crop Gene Exploration and Utilization in Southwest China, Sichuan Agricultural University, 611130 Wenjiang, Chengdu, China. [13]State Key Laboratory of Crop Genetics & Germplasm Enhancement and Utilization, Jiangsu Collaborative Innovation Center for Modern Crop Production, Nanjing Agricultural University, No.1 Weigang, 210095 Nanjing, Jiangsu, China. [14]Université Paris Cité, Institute of Plant Sciences Paris-Saclay (IPS2), F-75006 Paris, France. [15]Université Paris-Saclay, CNRS, INRAE, Univ Evry, Institute of Plant Sciences Paris-Saclay (IPS2), 91405 Orsay, France. [16]These authors contributed equally: Zijuan Li, Yuyun Zhang, Ci-Hang Ding, Yan Chen, Haoyu Wang. ✉e-mail: moussa.benhamed@universite-paris-saclay.fr; zc_dong@gzhu.edu.cn; jygou@cau.edu.cn; zhangyijing@fudan.edu.cn

