## [Peer Review File · Nature Communications]

LHP1-mediated epigenetic buffering of subgenome diversity and defense responses confers genome plasticity and adaptability in allopolyploid wheatREVIEWER COMMENTS

Reviewer #1 (Remarks to the Author):

In this study, Li et al., explore transcriptional regulation of subgenomes in wheat and find that LHP1, a member of the PcGs in plants, preferentially targets non-triad genes. By generating *lhp1* mutants, they show that LHP1 is important for deposition of H3K27me3 in certain regions and loss of LHP1 and H3K27me3 leads to transcriptional derepression of target genes. Some of these genes are related to flowering (*lhp1* deletion phenotype is early flowering and dwarfism) and defense against pathogens. A similar set of genes is upregulated in resistant wheat cultivars upon stripe rust infection and *lhp1* mutants in the susceptible cultivar JW1 are more resistant to stripe rust infection than wild type plants. This would suggest that resistance is not due to genetic variation but transcriptional variability.

This study addresses a very interesting question, the transcriptional regulation of subgenomes in polyploid plants. The combination of ChIP-seq, RNA-seq, phenotyping and computational analyses is nicely presented and highlights some new findings about the role of “epigenetic” mechanisms to regulate genome diversity. I have a few comments that I think need to be addressed before this study can be published.

Comments:

The authors state that N-N-N genes are non-triad genes. Technically, they do exist in all three genomes, just in varying copy numbers, so I am not sure if non-triad is the right term here.

How does the copy number affect downstream data analysis? How different are the different copies? Can they be differentiated during analysis and mapping? Do you see a difference in genes that have many copies vs just a few copies? How does expression of the different copies change? Is it one or just a few or all of them? Same for H3K27me3 enrichment.

How many replicates were used for each experiment (ChIP-seq, RNA-seq)? All experiments should be performed in replicates.

There also seems to be a strong enrichment in N-0-N genes? Did you have a closer look at those?

LHP1 ChIPs were performed in wheat protoplasts, while H3K27me3 ChIPs were performed in leaves. How similar are H3K27me3 patterns between these two conditions?

Could the increased resistance to stripe rust in the *lhp1* mutants be due to the developmental defect rather than expression of defense related genes?

Figure 2c: where do you find more H3K27me3?

Figure 2c+d: I think the labels for colors are switched.

Figure 5e: Another panel showing *lhp1* mutant RNA would be helpful.

Lane 156: What do you mean by “evolutionary rate of a protein”? And shouldn't it be gene?

Line 187: Should this be “natural” variation instead of “neutral” variation?

The methods part on spike-in could be a little extended to clarify the procedure.

The RNA-seq cut-offs are not very stringent, does the p-value include some kind of correction for multiple comparisons (false discovery rate?)?

Reviewer #2 (Remarks to the Author):

Review of “LHP1 is an epigenetic buffer of subgenome diversity and defense responses in allopolyploid wheat” Li et al.

The paper presents a noteworthy and compelling story that synthesizes several important disciplines: polyploidy, subgenome dominance, epigenome regulation, pathogen resistance, and evolutionary genetics. It is likely to be of broad interest. The authors show that, in genes shared by subgenomes, LHP1 promotes H3K37me3 and associated silencing, especially in defense-related genes. They show that these genes exhibit elevated genetic variation, which may reflect differences in mutation rate and environmentally-dependent selection. They show that under pathogen stress that these genes are upregulated and experience reduced H3K37me3 and that loss of LHP1-mediated silencing leads to increased pathogen resistance. Taken together, the paper does a nice job of zooming in on an interesting system with well-done experiments while also expanding to consider the broader implications for understanding relationships between epigenome regulation and evolutionary processes.

I am worried some readers may get lost in some of the specific terms/phrases. Some semantics should be sorted out in the introduction and throughout the paper to help the reader.

L 69: “ internal stress and may result in genetic conflicts” these (internal stress, genetic conflicts) may need to be defined. L75: “Buffering systems” should be explicitly defined prior to referencing it here. L76- what are “unknown phenotypes”? L77-78: “stabilizing expression” in what dimension? Between environments (ie. reducing plasticity in expression)? Between subgenomes (ie. reducing subgenome dominance in expression)? 81: “Epigenetic diversity” - does this mean differences at the same locus between subgenomes, or diversity between genotypes?

Other terms that would be good to define as plainly and clearly as possible “subgenome diversity”, “captured variations”, “polyploidy promoted diversity”, while I think I ultimately understood what the authors meant by these, it wasn't immediately clear when they were first mentioned in the paper.

L121: Define “N” in the main text

L133 (Fig 1.h) It might be worth briefly mentioning the nature of these data in the main text. What datasets were used?

Fig 2d, 3a,b,c (maybe 5a) - can the “enrichment” values be explained more - are values below 1 indicating less than expected? If these are fractions (ie. obs/exp) it may be advised to log transform the enrichment as it is in other plots (e.g. 5b)

The authors could consider adding, to Figure 3 and associated results, for example, analyses of non-synonymous(Ns) vs synonymous(S) variation to better parse the selective pressure and/or mutation rate differences between genes (e.g. Ns/S as measure of selection and S/b.p. to address mutation rate).

While interesting, I am not sure the section in the discussion about cancer is particularly effective.

I believe the paper has already demonstrated a lot and is quite compelling, but if I could make one suggestion for this or future work it would be to look at H3K4me1 in the wildtype/mutants and +- infection. A possible hypothesis to test would be that H3K4me1 will increase because it is a mark of active expression, but more importantly, H3K4me1 is the putative target of DNA repair proteins in plants (<https://www.biorxiv.org/content/10.1101/2022.05.28.493846v3.abstract>). This could be relevant to address one of the insightful points of the paper regarding the mechanistic relevance of the system for evolutionary patterns and promoting genetic variation in interesting ways. Moreover, the ultimate test would be to follow up with mutation accumulation experiments in wildtype and LHP1 mutants to compare mutation rates at loci targeted by LHP1.

I think I like Fig 7, but what do the axes of the bottom panel represent?

Signed,
Grey Monroe

We thank the reviewers for the constructive comments in helping us improve our manuscript. Please refer to the point-to-point response listed below:

REVIEWER COMMENTS

Reviewer #1 (Remarks to the Author):

In this study, Li et al., explore transcriptional regulation of subgenomes in wheat and find that LHP1, a member of the PcGs in plants, preferentially targets non-triad genes. By generating *lhp1* mutants, they show that LHP1 is important for deposition of H3K27me3 in certain regions and loss of LHP1 and H3K27me3 leads to transcriptional derepression of target genes. Some of these genes are related to flowering (*lhp1* deletion phenotype is early flowering and dwarfism) and defense against pathogens. A similar set of genes is upregulated in resistant wheat cultivars upon stripe rust infection and *lhp1* mutants in the susceptible cultivar JW1 are more resistant to stripe rust infection than wild type plants.

This would suggest that resistance is not due to genetic variation but transcriptional variability.

This study addresses a very interesting question, the transcriptional regulation of subgenomes in polyploid plants. The combination of ChIP-seq, RNA-seq, phenotyping and computational analyses is nicely presented and highlights some new findings about the role of “epigenetic” mechanisms to regulate genome diversity. I have a few comments that I think need to be addressed before this study can be published.

We thank the reviewer for the summary and the constructive comments on our manuscript.

Comments:

1. The authors state that N-N-N genes are non-triad genes. Technically, they do exist in all three genomes, just in varying copy numbers, so I am not sure if non-triad is the right term here.

Response : We thank the reviewer for pointing this out. We adjusted the term “triad genes” to “single copy triads (sc-triads)”, and “non-triad genes” to “non-sc-homoeologs”.

2. How does the copy number affect downstream data analysis? How different are the different copies? Can they be differentiated during analysis and mapping? Do you see a difference in genes that have many copies vs just a few copies? How does expression of the different copies change? Is it one or just a few or all of them? Same for H3K27me3 enrichment.

Response :

- 1) Can they be differentiated during analysis and mapping?

We performed the following analysis demonstrating that these genes could be differentiated. For each homolog or paralog from one subgenome belonging to one orthogroup as detected by orthofinder, the gene sequence is extracted using the sliding-window technique (window size = 150 bp; shift size = 1 bp). Each 150 bp sequence is compared to other genes in the same group for each subgenome. None of the multi-copy gene have 150 bp exactly the same with other

genes of the same group. Therefore, the multi-copy genes from one subgenome are theoretically distinguishable by sequencing the paired-end 150 bp read.

2) Do you see a difference in genes that have many copies vs just a few copies?

We categorized the homoeologs based on the copy number, and measured the enrichment for LHP1 targets (following figure). The higher the copy number, the higher the enrichment degree. We added this result to the revised manuscript (Supplemental Fig. S2)

3) How does expression of the different copies change? Same for H3K27me3 enrichment.

We observed that the higher the copy number, the higher the expression divergence and the higher enrichment of downregulated H3K27me3 in *lhp1* mutant (the following figures).

The higher the copy number, the higher the expression divergence.

The higher the copy number, the higher enrichment of downregulated H3K27me3 in *lhp1* mutant

4) Is it one or just a few or all of them?

Genes in each orthogroup were divided into four quantiles based on gene expression level (FPKM), low: <1; median: 1-3; high: 3-6; very high: >6. Orthogroups with only one gene at one level were defined as orthogroups having only one copy differed from the others. The ratios of such orthogroups are 12.72%, 12.25% and 12.94% for A, B and D subgenomes.

3. How many replicates were used for each experiment (ChIP-seq, RNA-seq)? All experiments should be performed in replicates.

Response : We thank the reviewer for pointing this out. We have two CRISPR-edited lines, both were used for generating ChIP-seq and RNA-seq data, and the results are consistent between these two mutant lines (Fig. S6-S9). In the revised manuscript, we added replicates for all data generated from wild type, including RNA-seq before and after pathogen treatment, H3K27me3 ChIP-seq after pathogen inoculation, and ChIP-seq of LHP1 binding loci in leaf protoplast (Table S2).

4. There also seems to be a strong enrichment in N-0-N genes? Did you have a closer look at those?

Response : We thank the reviewer for pointing this out. LHP1 binding is top enriched for homoeologs absent in one subgenome and multiple copies in other subgenome(s), while N-N-N is not among the top enriched groups. The high significance of N-N-N is mostly due to the large number of N-N-N homoeologs. All these top enriched groups and N-N-N preferentially involved in defense responses.

5. LHP1 ChIPs were performed in wheat protoplasts, while H3K27me3 ChIPs were performed in leaves. How similar are H3K27me3 patterns between these two conditions?

Response : We thank the reviewer for pointing this out. We performed H3K27me3 ChIP in leaf protoplast, which showed highly consistent pattern compared to LHP1 binding (Table S2).

6. Could the increased resistance to stripe rust in the *lhp1* mutants be due to the developmental defect rather than expression of defense related genes?

Response : We thank the reviewer for pointing out this possibility. We cannot rule out the possibility that developmental defects may affect defense responses, but the resistance phenotype is at least in part due to direct repression of defense-related genes by LHP1. Some well-known defense genes, including *PRI*, *CERK1*, *WRKY61*, *Stb16q*, are direct targets of LHP1-mediated H3K27me3. Their expressions are significantly induced in *lhp1* mutants, suggesting that these genes are repressed by LHP1 under normal conditions (Fig. 4). We also demonstrated that LHP1 targets and represses the expression of the recently reported four subgenome-diversified gene clusters mediating the synthesis of defense-related metabolites (Fig. S4). Furthermore, we detected apparent increase of ROS in *lhp1* mutants, a typical marker of defense responses (Fig. 6). Thus, LHP1 likely represent an epigenetic buffer system that controls the switch between growth and defense.

7. Figure 2c: where do you find more H3K27me3?

Figure 2c+d: I think the labels for colors are switched

Response: We apologize for the misunderstanding due to color usage. In the previous version, orange dots represent regions with reduced H3K27me3 (M value < 0). In the revised version, we used blue dots to represent repression. Figure 2d represents enrichment of homeologous groups in genes with differential H3K27me3 changes (M value) in *Talhp1-abd*. The color of dots represents enrichment score, orange represents enrichment and blue represents depletion.

8. Figure 5e: Another panel showing *lhp1* mutant RNA would be helpful.

Response : We thank the reviewer for pointing this out. We added the panel showing RNA-seq data in *lhp1* mutant.

9. Lane 156: What do you mean by “evolutionary rate of a protein”? And shouldn’t it be gene?

Response : We thank the reviewer for pointing this out. This statement follows a previous report focusing on protein sequence evolution (doi: 10.1038/nrg3950), and protein is used here to emphasize coding genes. We now revised it to “evolutionary rate of a gene” to facilitate the understanding (lines 168 and 170).

10. Line 187: Should this be “natural” variation instead of “neutral” variation?

Response : We thank the reviewer for pointing this out. By comparing the genetic diversity already present in diploid progenitors (diversified across subgenomes) and the variation between polyploid wheat of the same subgenome, we found that LHP1 binding highly enriched in variable subgenome regions in progenitors which are further diverge after allopolyploidization, possibly selectively neutral due to genetic drift. Thus, we concluded that “These findings and evidence imply LHP1 repression protects and potentially promotes the accumulation of neutral or weakly deleterious variations across subgenomes.” Please refer to lines 200-202.

11. The methods part on spike-in could be a little extended to clarify the procedure.

Response : We added the description of spike-in procedure in Methods line 447-454.

12. The RNA-seq cut-offs are not very stringent, does the p-value include some kind of correction for multiple comparisons (false discovery rate?)?

Response: All RNA-seq cut-offs are changed to false discovery rate (Benjamini-Hochberg corrected). Please refer to lines 529-530 in Methods.

Reviewer #2 (Remarks to the Author):

Review of “LHP1 is an epigenetic buffer of subgenome diversity and defense responses in allopolyploid wheat” Li et al.

The paper presents a noteworthy and compelling story that synthesizes several important disciplines: polyploidy, subgenome dominance, epigenome regulation, pathogen resistance, and evolutionary genetics. It is likely to be of broad interest. The authors show that, in genes shared by subgenomes, LHP1 promotes H3K37me3 and associated silencing, especially in defense-related genes. They show that these genes exhibit elevated genetic variation, which may reflect differences in mutation rate and environmentally-dependent selection. They show that under pathogen stress that these genes are upregulated and experience reduced H3K37me3 and that loss of LHP1-mediated silencing leads to increased pathogen resistance. Taken together, the paper does a nice job of zooming in on an interesting system with well-done experiments while also expanding to consider the broader implications for understanding relationships between epigenome regulation and evolutionary processes.

I am worried some readers may get lost in some of the specific terms/phrases. Some semantics should be sorted out in the introduction and throughout the paper to help the reader.

We thank the reviewer for the summary and the constructive comments on our manuscript.

1. L 69: “ internal stress and may result in genetic conflicts” these (internal stress, genetic conflicts) may need to be defined.

Response: We included detailed description about the “internal stress” and “genetic conflict” introduced by polyploidization. Please refer to lines 70-76, “However, the convergence of different genomes may not necessarily result in heterosis. Instead, the genetic heterogeneity between subgenomes (hereafter referred to as ‘subgenome diversity’) may result in genetic conflicts. Typical examples include competition between parental genomes and the rapid loss or repression of homoeologous gene copies (biased fractionation or repression)”.

2. L75: “Buffering systems” should be explicitly defined prior to referencing it here.

Response: We explained the “buffering system” in the revised manuscript. Please refer to lines 79-82. “Genetic diversity and environmental stimuli respectively represent internal and external stresses affecting organisms. Phenotypes must be robust in response to internal and external changes, which requires a buffering system to ensure normal development (‘canalization’)”

3. L76- what are “unknown phenotypes”?

Response: We thank the reviewer for pointing this out. The “unknown phenotypes” is changed to “uncovered phenotypes” (line 83).

4. L77-78: “stabilizing expression” in what dimension? Between environments (ie. reducing plasticity in expression)? Between subgenomes (ie. reducing subgenome dominance in expression)?

Response: We thank the reviewer for pointing this out. We added the dimension, and revised the sentence to “In eukaryotes, chromatin and the associated epigenetic mechanisms represent a typical buffering system that stabilizes transcription and cellular homeostasis against internal and external changes, while also reprogramming the transcriptome in response to developmental or environmental cues” (lines 84-87).

5. “Epigenetic diversity” - does this mean differences at the same locus between subgenomes, or diversity between genotypes?

Response: We thank the reviewer for pointing this out. We revised the sentence to “Epigenetic diversity **across subgenomes** and the flexible interaction with the transcriptional machinery may provide polyploid wheat with a selective advantage” (line 89).

Other terms that would be good to define as plainly and clearly as possible

6. “subgenome diversity”, “captured variations”, “polyploidy promoted diversity”, while I think I ultimately understood what the authors meant by these, it wasn't immediately clear when they were first mentioned in the paper.

Response: We thank the reviewer for pointing this out. We now added details when these terms were first mentioned in the paper. Please refer to line 71 “the genetic heterogeneity between subgenomes (hereafter referred to as ‘subgenome diversity’)”, line 175 “captured from diploid progenitors (i.e., variations already present across diploids)”, and line 179 “polyploidy

promoted diversity” is changed to “polyploidy-generated diversity”. Please refer to lines 175-184.

7. L121: Define “N” in the main text

Response: We thank the reviewer for pointing this out. We revised the sentence to “especially the genes with an N:N:N correspondence across subgenomes (N indicates a minimum of one additional paralog per respective subgenome)” (lines 130-132).

8. L133 (Fig 1.h) It might be worth briefly mentioning the nature of these data in the main text. What datasets were used?

Response: We thank the reviewer for pointing this out. We revised the sentence to “A phylostratigraphic analysis, which clustered the genes of common wheat based on their orthologous relationships with nine other species (see Methods), revealed an apparent burst of H3K27me3 sites in Triticeae” (lines 142-143).

9. Fig 2d, 3a,b,c (maybe 5a) - can the “enrichment” values be explained more - are values below 1 indicating less than expected? If these are fractions (ie. obs/exp) it may be advised to log transform the enrichment as it is in other plots (e.g. 5b)

Response: All enrichment values were log transformed.

10. The authors could consider adding, to Figure 3 and associated results, for example, analyses of non-synonymous(Ns) vs synonymous(S) variation to better parse the selective pressure and/or mutation rate differences between genes (e.g. Ns/S as measure of selection and S/b.p. to address mutation rate).

Response: We thank the reviewer for pointing this out. dN/dS analysis are generally applied to 1:1 orthologous gene to test the selection. Here, LHP1 targets are mostly non 1:1 homeologs, and thus we employed population data for measuring the selection.

11. While interesting, I am not sure the section in the discussion about cancer is particularly effective.

Response: We were initially hesitant whether to include this part. However, given that polyploidy is a hallmark of cancer¹, high genetic heterogeneity is one major factor contributing to the high plasticity of cancer cells¹, and the oncogenic role of PcGs (LHP1 as the major component) is under increased scrutiny for their cancer therapeutic value^{2,3}, it will be interesting to see whether our findings regarding the translation of genetic diversity to environmental plasticity and adaptive evolution via the buffering effect of PcGs are applicable to cancer cells. Since ‘Nature Communications’ is a general journal and there is high likely to be read by researchers in other fields, it will be of great significance if the findings from polyploid plants are verified in cancer research.

References

- 1 Meacham, C. E. & Morrison, S. J. Tumour heterogeneity and cancer cell plasticity. *Nature* **501**, 328-337, doi:10.1038/nature12624 (2013).
- 2 Piunti, A. & Shilatifard, A. The roles of Polycomb repressive complexes in mammalian development and cancer. *Nat Rev Mol Cell Biol* **22**, 326-345, doi:10.1038/s41580-021-00341-1 (2021).

3 Kim, K. H. & Roberts, C. W. Targeting EZH2 in cancer. *Nat Med* **22**, 128-134, doi:10.1038/nm.4036 (2016).

12. I believe the paper has already demonstrated a lot and is quite compelling, but if I could make one suggestion for this or future work it would be to look at H3K4me1 in the wildtype/mutants and +/- infection. A possible hypothesis to test would be that H3K4me1 will increase because it is a mark of active expression, but more importantly, H3K4me1 is the putative target of DNA repair proteins in plants (<https://www.biorxiv.org/content/10.1101/2022.05.28.493846v3.abstract>). This could be relevant to address one of the insightful points of the paper regarding the mechanistic relevance of the system for evolutionary patterns and promoting genetic variation in interesting ways. Moreover, the ultimate test would be to follow up with mutation accumulation experiments in wildtype and LHP1 mutants to compare mutation rates at loci targeted by LHP1.

Response: We thank the author for these insightful suggestions, which are interesting points for subsequent works that follow.

13. I think I like Fig 7, but what do the axes of the bottom panel represent?

Response: We thank the reviewer for pointing this out. The axes represent the evolutionary direction, we included a red arrow and label in the revised figure.

Signed,
Grey Monroe

Thank you for your suggestions and comments, it is a pleasure to get acquainted in this way.

Reviewer #3 (Remarks to the Author):

Wheat is an important crop with polyploid nature and an ideal system to study the epigenetics variation during the polyploidization process. Understanding the epigenetics changes during the polyploidization in regulating the developmental process, response to stress and adaptation is crucial for wheat fundamental scientific questions and also benefit for the molecular breeding of novel wheat cultivars with better adaptation and stress tolerance. The manuscript by Li et al. identified the Like Heterochromatin Protein 1 (LHP1) as an epigenetic buffer of subgenome diversity and defense responses in allopolyploid wheat. The critical roles of LHP1 in regulating flowering, plant architecture (dwarfness), and disease resistance were demonstrated. The finding links the epigenetics regulation to important wheat agronomic traits that is valuable for the science society.

I have a few major concerns for the current manuscript.

We thank the reviewer for the summary and the constructive comments on our manuscript.

1. The authors identified LHP1 as a master repressor of subgenome-diversified genes. LHP1 was the primary trans-factor regulating non-triad genes in hexaploid wheat. Is this also true in tetraploid wheat durum? They observed a LHP1 repression of subgenome-diversified gene

activity and proposed LHP1 may influence evolutionary processes. It is necessary to compare the LHP1 effects between wheat in tetraploid and hexaploid level.

Response : We thank the reviewer for pointing this out. The transgenic technology is still unavailable for tetraploid wheat, we don't have *lhp1* mutant in tetraploid wheat. Since LHP1 binding is highly consistent with H3K27me3 mark shown in this study (Fig. 1f) and reported elsewhere (doi: 10.1371/journal.pgen.0030086, doi: 10.1371/journal.pgen.1005771), we compared the H3K27me3 level between 1:1 orthologous gene and non-1:1-orthologous gene in tetraploid wheat and other polyploid crops. Enrichment of H3K27me3 in non-1:1-orthologous genes were detected for all these polyploid species. Please refer to lines 308-312 and Fig. S5.

Supplemental Figure 5. H3K27me3 preferentially targeted non 1:1 homoeologous genes in polyploidy species.

2. Global decreasing in H3K27me3 levels surrounding non-triad genes in the LHP1 triple mutants. What is the H3K27me3 patterns in the LHP1 single and double mutants?

Response : We detected the global H3K27me3 level in *lhp1* triple, single and double mutants via western blot with specific antibody against H3K27me3. The triple mutant has apparent H3K27me3 reduction compared to single and double mutants (lines 152-154 and Fig. S3).

3. L228-232: Why the Yr16 gene stock was used for this analysis? Is there confident evidence that JW1 contains Yr16 since Yr16 was still not cloned? Is Yr16 resistant to the Pst isolate CYR32 used for inoculation?

Response: We apologize for the misunderstanding caused by the unclear description. We detected apparent enrichment of defense related genes targeted by LHP1-mediated H3K27me3, whose expressions are generally upregulated in *lhp1* mutants. In order to predict the type of pathogen response preferentially regulated by LHP1, “we designed a statistical approach that integrates publicly available transcriptomic data. The underlying principle of this approach is that effectors in the same pathway likely trigger overlapping downstream cascades, which may be reflected by the similarity of the transcriptomic changes. For example, if mutations in two genes trigger similar differential expression patterns, it is likely that these two genes are functionally relevant. We obtained the transcriptomes of samples resistant and susceptible to various pathogens. For each pathogen, the transcriptomic changes were compared between the resistant and susceptible samples. The correlation between the transcriptomic changes revealed by these pairwise comparisons and *Talhp1-abd* mutant-induced changes was assessed (Fig. 5a). The sample with transcriptomic changes that were most enriched with *Talhp1-abd* mutant-induced changes was identified as a near isogenic line (FLW29) containing a locus mediating the resistance to *Puccinia striiformis* f. sp. *tritici* (*Pst*), which causes one of the most widely destructive cereal diseases (i.e., stripe rust). In other words, the lack of *LHP1* triggered similar transcriptomic changes as the introduction of a *Pst* resistant locus”. Please refer to lines 230-245.

4. It is interesting to see the *Talhp1-abd* triple mutant exhibits stripe rust resistance in the seedling stage. The authors observed an increased expression of PR gene in the LHP1 knockout plant, it is necessary to test the adult plant resistance (APR) of stripe rust resistance. The increasing PR expression usually results broad spectrum resistance to multiple pathogens. I would suggest the authors to perform additional tests on other biotrophic fungi such as leaf rust and powdery mildew.

Response: We tested the resistance to different races of stripe rust, and detected apparent resistance to three common races in *Talhp1-abd* triple mutant. The resistance to different races suggesting a broad-spectrum stripe rust resistance of *LHP1* deprivation.

5. Minor: L47, L68-69: “the merger of different genomes may result in genomic conflicts”.

This is not always true. Ployploidization in Triticeae family usually results hybrid vigor and better adaptation as evidenced in allohexaploid wheat.

Response : We thank the reviewer for pointing this out. We revised the statement to “However, the convergence of different genomes may not necessarily result in heterosis. Instead, the genetic heterogeneity between subgenomes (hereafter referred to as ‘subgenome diversity’) may result in genetic conflicts. Typical examples include competition between parental genomes and the rapid loss or repression of homoeologous gene copies (biased fractionation or repression). Persistent aneuploidy and decreased fertility are generally associated with nascent allohexaploid wheat”. Please refer to lines 70-76.

6. L50: LHP1: Like heterochromatin protein 1 (LHP1).

Response : We thank the reviewer for pointing this out. The full name is now used in the Abstract (line 51).

REVIEWERS' COMMENTS

Reviewer #2 (Remarks to the Author):

I remain enthusiastic about this work. The authors have done a great job addressing my previous comments.

Reviewer #3 (Remarks to the Author):

The authors revised their manuscript based on the comments of the reviewers. The content and quality of the revision is significantly improved. I was satisfied with most of the responses of the comments. However, there are still two questions need to be addressed.

Question 1:

The authors responded that “The transgenic technology is still unavailable for tetraploid wheat, we don't have lhp1 mutant in tetraploid wheat”. This is not true. The coauthor of this manuscript developed transgenic Kronos transgenic wheat in the publication of Chen et al. 2020 Nature Commun 11:6266 (All the mutants and transgenic wheat used in this work were in *Triticum turgidum* (tetraploid, AABB genome) Kronos ecotype background.).

Question 4:

The authors tested the stripe rust reaction of Talhp1-abd to only three CYR races. That could not be concluded as “broad-spectrum resistance”. Are the reactions of Talhp1-abd to CYR32, CYR33 and CYR34 shown in the picture at the seedling stage? Or at the adult plant stage?

Reviewer #3 (Remarks to the Author):

The authors revised their manuscript based on the comments of the reviewers. The content and quality of the revision is significantly improved. I was satisfied with most of the responses of the comments. However, there are still two questions need to be addressed.

Question 1:

The authors responded that “The transgenic technology is still unavailable for tetraploid wheat, we don’t have *lhp1* mutant in tetraploid wheat”. This is not true. The coauthor of this manuscript developed transgenic Kronos transgenic wheat in the publication of Chen et al. 2020 Nature Commun 11:6266 (All the mutants and transgenic wheat used in this work were in *Triticum turgidum* (tetraploid, AABB genome) Kronos ecotype background.).

Response: We thank the reviewer for the reminder. Testing whether deprivation of *lhp1* in tetraploid wheat has similar epigenomic and resistance phenotypes to hexaploid wheat is definitely an interesting issue for subsequent studies. In the current study, we demonstrate that H3K27me3 is highly enriched surrounding subgenome-diversified genes in polyploid plants, and is most prominent in tetraploid and hexaploid wheat, indicating that PcG-mediated epigenetic control of subgenome diversity is likely a common mechanism among polyploids. (Fig. S5 and lines 307-311).

Question 4:

The authors tested the stripe rust reaction of *Talhp1-abd* to only three CYR races. That could not be concluded as “broad-spectrum resistance”. Are the reactions of *Talhp1-abd* to CYR32, CYR33 and CYR34 shown in the picture at the seedling stage? Or at the adult plant stage?

Response: We thank the reviewer for pointing this out. We detected apparent resistance in *lhp1-abd* seedlings. We have specified the developmental stage for inoculation (line 58 and line 264). Since LHP1 affects the developmental progress, more issues need to be considered when comparing resistance at the adult stage, such as growth time or development period consistency, the influence of the environment, etc. Searching for stripe rust-resistant *lhp1* mutant without yield loss in the field is the target of our follow-up investigations.